# Accurate and rapid antibiotic susceptibility testing using a machine learning-assisted nanomotion technology platform

Alexander Sturm [1] ✉, Grzegorz Jóźwiak[1], Marta Pla Verge [1], Laura Munch[1], Gino Cathomen [1], Anthony Vocat[1], Amanda Luraschi-Eggemann[1], Clara Orlando[1], Katja Fromm [1], Eric Delarze[1], Michał Świątkowski[1], Grzegorz Wielgoszewski [1], Roxana M. Totu [1], María García-Castillo[2], Alexandre Delfino[3], Florian Tagini [3], Sandor Kasas[4,5], Cornelia Lass-Flörl [6], Ronald Gstir[6], Rafael Cantón [2,7], Gilbert Greub [3,8] & Danuta Cichocka[1,8]

Antimicrobial resistance (AMR) is a major public health threat, reducing treatment options for infected patients. AMR is promoted by a lack of access to rapid antibiotic susceptibility tests (ASTs). Accelerated ASTs can identify effective antibiotics for treatment in a timely and informed manner. We describe a rapid growth-independent phenotypic AST that uses a nanomotion technology platform to measure bacterial vibrations. Machine learning techniques are applied to analyze a large dataset encompassing 2762 individual nanomotion recordings from 1180 spiked positive blood culture samples covering 364 *Escherichia coli* and *Klebsiella pneumoniae* isolates exposed to cephalosporins and fluoroquinolones. The training performances of the different classification models achieve between 90.5 and 100% accuracy. Independent testing of the AST on 223 strains, including in clinical setting, correctly predict susceptibility and resistance with accuracies between 89.5% and 98.9%. The study shows the potential of this nanomotion platform for future bacterial phenotype delineation.

Antimicrobial resistance (AMR) has become a significant threat to public health worldwide with almost five million AMR associated deaths in 2019[1]. AMR largely stems from excessive and improper antibiotic use in healthcare and animal husbandry[2–4]. Its proliferation, combined with a lack of industry investment in antibiotic research[5] and a low pass rate for antimicrobials in clinical trials, has caused an alarming decrease in treatment options available for infected patients[6,7]. Consequently, the clinical usage of last-resort antibiotics

has expanded, but this comes at the cost of promoting AMR spread. AMR diagnostic strategies such as antibiotic susceptibility testing (AST) help provide clinicians with sufficient data to reach informed decisions and promote the administration of narrow-spectrum antibiotics[4]. In turn, treatment de-escalation is vital in reducing the spread of AMR while maintaining therapeutic efficacy for the patient, shortening hospital stay durations, and reducing healthcare-associated costs. Among the most critical and prevalent threats are

[1]Resistell AG, Hofackerstrasse 40, 4132 Muttenz, Switzerland. [2]Hospital Universitario Ramón y Cajal, Instituto Ramón y Cajal de Investigación Sanitaria (IRYCIS), Carretera de Colmenar Km 9,1, 28034 Madrid, Spain. [3]Institute of Microbiology, Lausanne University Hospital (CHUV) & University of Lausanne (UNIL), 1011 Lausanne, Switzerland. [4]Laboratory of Biological Electron Microscopy (LBEM), École Polytechnique Fédérale de Lausanne (EPFL) and University of Lausanne (UNIL), 1015 Lausanne, Switzerland. [5]Centre Universitaire Romand de Médecine Légale (UFAM) & Université de Lausanne (UNIL), 1015 Lausanne, Switzerland. [6]Institut für Hygiene und Medizinische Mikrobiologie, Medizinische Universität Innsbruck, Schöpfstraße 41, 6020 Innsbruck, Austria. [7]CIBER de Enfermedades Infecciosas (CIBERINFEC). Instituto de Salud Carlos III. Sinesio Delgado 4, 28029 Madrid, Spain. [8]These authors contributed equally: Gilbert Greub, Danuta Cichocka. ✉e-mail: alex.sturm@resistell.com

*Enterobacterales* resistant to third-generation cephalosporins[8]. From this group, *Escherichia coli* and *Klebsiella pneumoniae* are the most frequently occurring Gram-negative pathogens in bloodstream infections (BSI) in Europe[9,10]. In the event of cephalosporin resistance, fluoroquinolones are commonly used. In cases of resistance to both classes, treatment is escalated to carbapenems or cephalosporin-β-lactamase inhibitor combinations.

Many current phenotypic AST methodologies rely on detecting growth metrics, such as doubling time or biomass change. This limits further technological development in minimizing the time to result (TTR), despite numerous recent innovations utilizing microfluidics, advanced sensor systems, and analysis pipelines of increasing sophistication[11–16]. All current workhorse instruments in European clinics rely on automated bacterial growth measurements[17–19] or disk diffusion assays (Kirby Bauer)[20]. These methods can offer next-day turnaround, at best, for fast-growing bacteria such as those frequently found in BSI. The TTR can be as high as one month for slow-growing bacteria such as *Mycobacterium tuberculosis*. This lag poses a significant obstacle to select the best treatment option for critically ill patients[21].

Nanomotion technology is based on atomic force microscopy (AFM)[22] and has been proposed as potential means of circumventing these limitations by measuring bacterial viability and response to antibiotics in a growth-independent manner[23–30]. Here, a functionalized cantilever oscillates in response to bacterial vibrations. Conditions that alter bacterial vibrations, such as drug exposure, modulate cantilever oscillations, and these changes can be detected, measured, and outputted using an optical read-out system[23,31]. Nanomotion technology can distinguish the differing responses of resistant and susceptible bacteria to antibiotic treatment and has already been employed to detect antibiotic susceptibility for several different microorganisms, including *Enterobacterales, Staphylococcus aureus, M. tuberculosis,* and *Candida albicans*[23,30,32–36]. However, these studies largely delivered a proof-of-concept for a nanomotion AST, using laboratory prototypes to study a few susceptible and resistant reference strains. Until now, antibiotic effects on bacteria were analysed by quantifying the cantilever position variance as a function of time. To address the diversity of strains encountered in the clinical setting with a wide range of minimal inhibitory concentrations (MICs) and various responses to a given antibiotic, the set of analysis tools of nanomotion signals needed to be expanded to develop a clinically relevant AST.

For that matter, we have developed an integrated nanomotion technology platform with both hardware and software components that demonstrates high test repeatability and reproducibility. This platform comprises the Phenotech device and cantilever sensors but also includes sample preparation, data acquisition at a frequency of 60 kHz, and advanced analysis of large datasets (Fig. 1a) available to the general microbiologist without prior knowledge of AFM. It is a robust, highly sensitive, and user-friendly technology. The cell attachment kit facilitates fast sample preparation from positive blood cultures (PBCs) and prevents bacterial detachment during the experiment. The biggest enhancement, however, is the development of an advanced mathematical analysis strategy for large datasets. Raw nanomotion signals initially appear random and require signal transformation for interpretation. One of the signal transformations employed in earlier studies was the comparative analysis of the variance over time[23]. This novel nanomotion technology platform employs supervised machine learning (ML) to develop classification models for different susceptibility phenotypes. To differentiate between clinical samples containing antibiotic-resistant or -susceptible bacteria, it extracts signal parameters (SP) from the power spectrum (PSD) over different time intervals within the nanomotion signal (Supplementary Information). A few informative SPs were selected from a pool of >100,000 to create a given classification model. Some examples of SPs include the variance ratio between time intervals, the slope derived from an exponential fit of the variance curve, the integral within a certain frequency range of the PSD, flicker noise, minima, maxima or slopes etc. in the PSD, and ratios derived of these measurements (Supplementary Information). Afterward, the models were cross-validated and tested on independent test datasets (Methods and Supplementary Fig. 1).

In the present study, we used the Phenotech nanomotion technology platform to generate and analyse a large dataset investigating the response of 352 *E. coli* and *K. pneumoniae* isolates (Supplementary Data 1) to four clinically important antibiotics (from two different classes) used to treat BSI: the fluoroquinolone ciprofloxacin (CIP) as well as the cephalosporins ceftriaxone (CRO), cefotaxime (CTX), and ceftazidime in combination with the β-lactamase inhibitor avibactam (CZA). We used this data to build specific machine-learning-generated classification AST models and compared their performance to existing reference growth-based methodologies. We demonstrate how increasing algorithm complexity to Pareto optimality[37] by increasing the number of inherent SPs until saturating performance. The final models for our Phenotech-based nanomotion AST achieved accuracies on independent test datasets ranging from 89.5% for CIP to 98.9% for CRO evaluating 4-h recordings and 93.0% for CZA based on 2-h recordings.

## Results

### Nanomotion AST experimental design

In contrast to classical methodologies, which typically use a set or gradient of antibiotic concentrations, we measured bacterial nanomotions only at a single antibiotic concentration above the clinical breakpoints to see the effect of the antibiotic fast[23,30,32,38]. For each experiment, a few hundred bacterial cells directly isolated from spiked positive blood cultures (PBCs) were attached to the cantilever (Fig. 1b, c and Supplementary Fig. 2). Bacterial nanomotions were recorded during incubation for 2 h with 50% LB broth (medium phase) and subsequently with 50% LB broth and an antibiotic (drug phase). In line with previous studies, upon adding the cephalosporin CRO, we observed decreased variance over time for the susceptible *E. coli* strain ATCC-25922 but sharply increased variance for the resistant strain BAA-2452 (Fig. 1d)[23,30]. Previous studies have measured mean medium phase/drug phase variance over short intervals and taken that ratio as the sole indicator for susceptibility[23,32,34]. This parameter can thus be considered a very simple but informative signal parameter (SP), i.e., a discrete mathematical value extracted from the signal or a part of the signal. Using the ratio between the variance from the drug and medium phase or parts of them as described in previous publications[23,32] did not suffice in separating different clinical isolates (Supplementary Fig. 3).

### Nanomotion recordings capture dose-based responses and strain diversity

Since nanomotion AST was conducted using a single antibiotic concentration, it was imperative to elucidate the concentration that best discriminated between susceptibility and resistance. Therefore, we performed several nanomotion recordings for both reference strains treated with CRO at concentrations ranging from sub-MIC to highly inhibitory concentrations. For both strains, we observed a sharp increase in signal variance after adding CRO at sub-MIC concentrations—an increase steeper than the variance in the absence of the drug. It is unclear whether this phenomenon is due to specific resistance mechanisms (e.g., extended-spectrum β-lactamases, pumps, target mutations, etc.) or stems from a common stress response that manages the global impact of the drug at sub-inhibitory concentrations[39–41]. Variance decreased or showed a lesser increase than medium phase controls when bacteria were treated with supra-MIC concentrations of CRO (Fig. 2a).

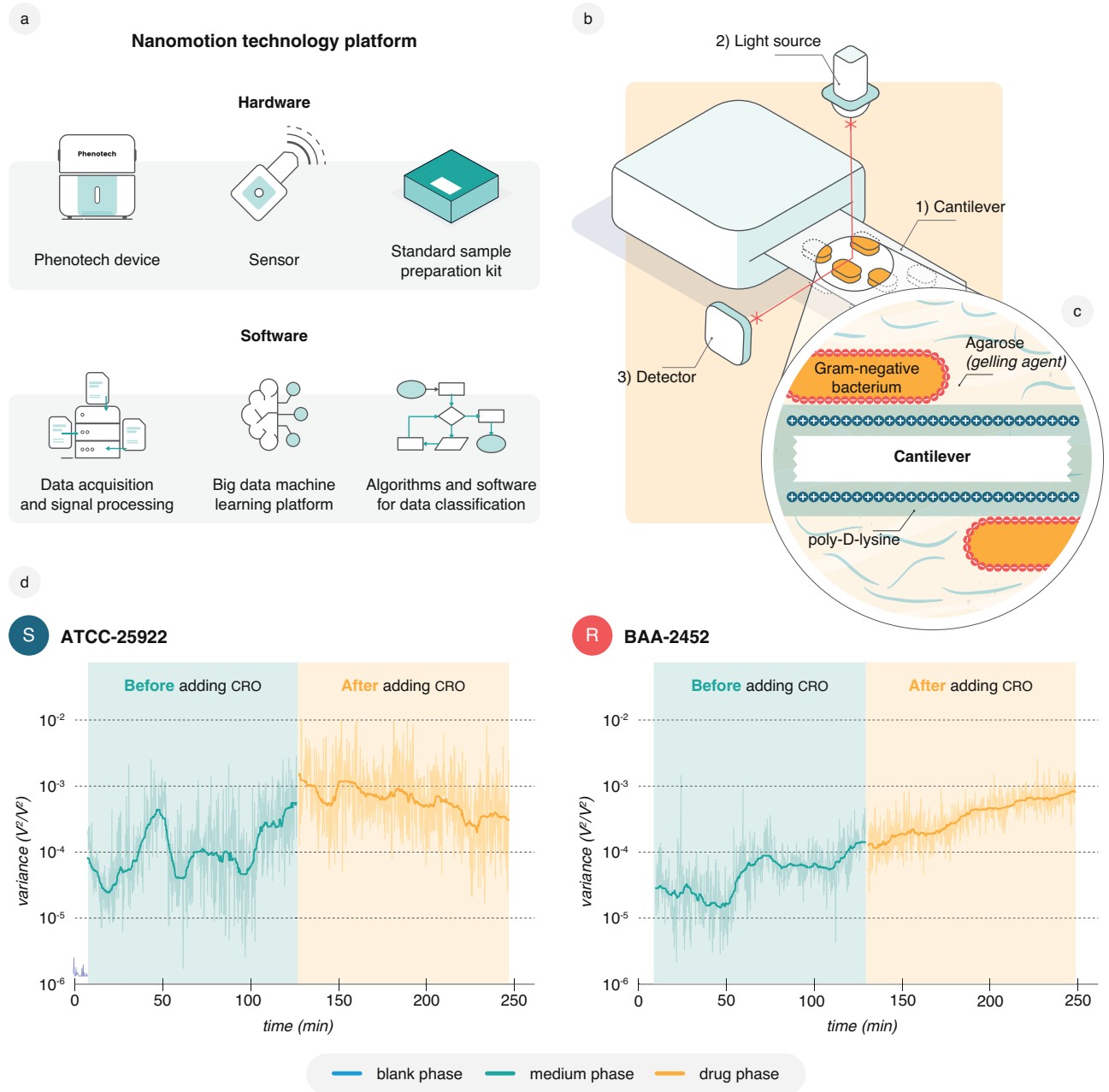

**Fig. 1 | Nanomotion detection and recording platform. a** Representation of the components of the nanomotion technology platform. **b** A representation of the nanomotion measurement setup with the (1) bacteria-loaded cantilever, (2) superluminescent light emitting diode (SLED) = light source, and (3) photo-detector. **c** Schematic illustrating Gram-negative bacteria attached to the cantilever. Prior to attachment, bacteria are dispersed in gelling agarose while the cantilever surface is functionalized using positively charged poly-D-lysine. The gelling agent proved beneficial for an even distribution and stability of the bacterial attachment. **d** Representative standard 4-h nanomotion recordings with a 2-h medium phase (50% LB medium) followed by a 2-h drug phase with 32 µg/ml CRO for the *E. coli* reference strains ATCC-25922 (S, susceptible) and BAA-2452 (R, resistant). These recordings form the basis for using nanomotion to conduct AST. This study contains 219 recordings of ATCC-25922 and 225 recordings of BAA-2452 exposed to 32 µg/ml CRO with similar results. Data are available in the source data file.

As a first attempt to find better descriptive SPs, we quantified the increase in signal variance during the 2-h CRO exposure period by fitting an exponential equation to the generated data points. This equation for the slope of the variance revealed that the distance between susceptible and resistant reference strains was at its greatest for CRO at a concentration of 32 µg/ml. Identifying an SP, such as the slope, enabled us to distinguish the nanomotion responses of both strains at several CRO concentrations (Fig. 2b). Dose-response testing using the SP slope as the measure was performed for all species-antibiotic combinations (Supplementary Fig. 4). The observation that ATCC-25922 exhibited similar slopes at both the MIC and media control supports nanomotion as a growth-independent method and suggests that cellular processes continue at the MIC as cells are not presumed to be dead at this concentration and still exhibit metabolic activity. This is in line with previous nanomotion observations[23].

Since CRO at 32 µg/ml showed the best discrimination for the two reference strains ATCC-25922 and BAA-2452, we continued recording nanomotion responses to CRO for different *E. coli* and *K. pneumoniae* strains at that concentration. The bacterial response to the CRO exhibited high variability in the slope of the variance curve among

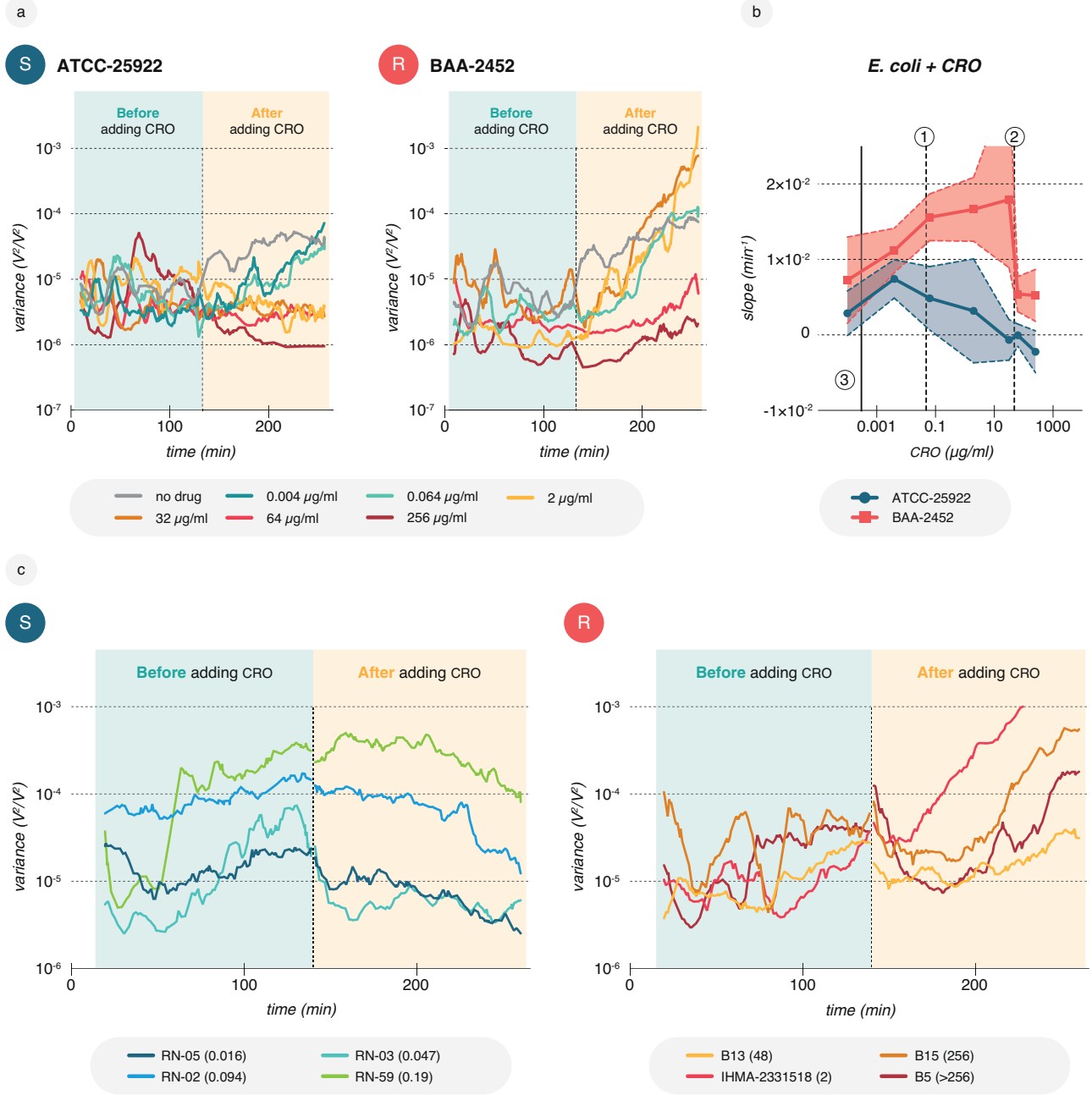

**Fig. 2 | Nanomotion signals vary depending on antibiotic concentration and strain variability. a** Representative nanomotion recordings for *E. coli* ATCC-25922 (S, MIC = 0.0048–0.064 µg/ml) and *E. coli* BAA-2452 (R, MIC = 48–64 µg/ml) cells when treated with different CRO concentrations. A dose-response pattern is evident when comparing the slopes of the variance during the drug phase. **b** CRO concentration-dependent slopes during drug phase calculated by the formula $\log(x) = \log(C) + at$, where $t$ is time, $a$ is the slope of the common logarithm of the variance trend, and $\log(C)$ is the intercept. (1) and (2) represent the MICs of ATCC-25922 (0.047 µg/ml) and BAA-2452 (48 µg/ml), respectively, while (3) represents the control without CRO. Shown are mean and SD of triplicates per concentration. **c** Nanomotion signal variability for four susceptible and four resistant *E. coli* isolates (additional combinations can be found in Supplementary Fig. 5). Shown are representative recordings for each isolate. Data are available in the source data file.

susceptible or resistant strains (Fig. 2c), suggesting a single empirically found SP would not suffice for extensive datasets comprising various clinical isolates with different drug response curves. Generalizing our nanomotion signal observations across several strains posed an obstacle. Additionally, the slope did not present a clear correlation to the MIC. For instance, IHMA-2331518 (2 µg/ml MIC) displayed a steep increase in the variance curve around the CRO-resistant breakpoint, whereas B5 with a much higher MIC of 256 µg/ml exhibited a more moderate increase (Fig. 2c). Similarly, B13 had a lower MIC than B5 but showed a less pronounced variance increase. Diverse responses were

also observed for *K. pneumoniae* and *E. coli* when treated with CIP or CTX. For example, the nanomotion signal of resistant strains could temporarily drop upon drug addition (Fig. 2c and Supplementary Fig. 5a–c). Consequently, we could not establish a clear trend between the MIC and SPs like the slope or simple statistical SPs such as the variance median or mean. Although the lack of correlation to MICs might appear counterintuitive, this may be explained by the induction of cellular stress leading to increased nanomotions below and around the MICs[39,42–44]. Distinct resistance mechanisms may also cause varied nanomotion responses among strains carrying different beta-

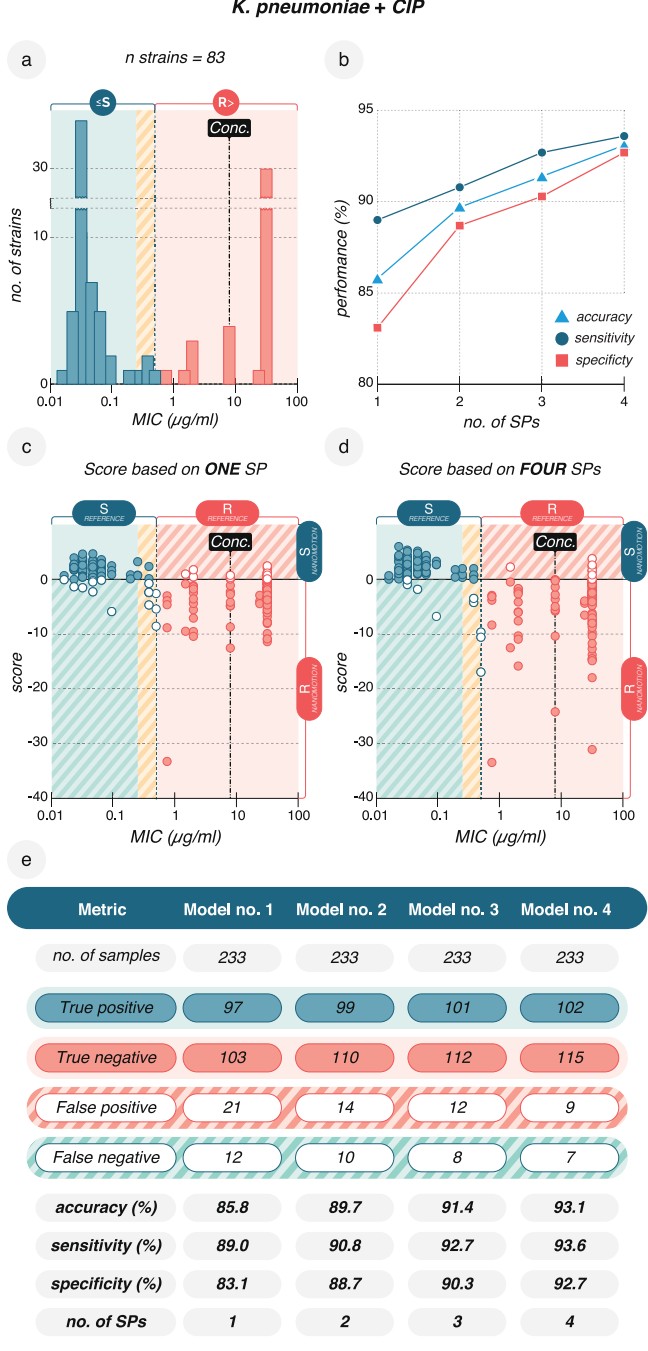

**Fig. 3 | Classification models with different numbers of signal parameters (SPs) differ in nanomotion AST performance for *K. pneumoniae* and CIP. a** The MIC distribution of the 83 *K. pneumoniae* isolates used to train classification models. The dotted line represents the border between the ≤S and R> classes, EUCAST (2022). *Conc.* (dashed line) indicates a CIP concentration of 4 μg/ml for nanomotion measurements. The yellow shaded area represents the "susceptible increased exposure category" combined with the S class (green) in nanomotion AST. **b** Classification model accuracy, sensitivity, and specificity improve with more SPs. **c** Classification according to nanomotion AST based on one SP. Each circle represents a single nanomotion measurement for which a score was calculated with the two classes defined as S > 0 > R. Closed circles show correctly classified measurements [True Positive (TP, correctly classified susceptible isolates), True Negative (TN, correctly classified resistant isolates)], and open circles show falsely classified measurements [False Positive (FP, falsely classified resistant isolates), False Negative (FN, falsely classified susceptible isolates)]. **d** Classification according to nanomotion AST based on four SPs. The number of falsely classified experiments decreased. **e** Classification model performance over 233 recordings improved as more SPs were introduced. Data are available in the source data file. Single SP values and scores are available in Supplementary Data 2.

and were benchmarked against MIC strip or broth micro-dilution. The ML algorithm autonomously selected a handful of the most relevant SPs, as expected primarily from the drug phase, containing information about antibiotic susceptibility irrespective of strain diversity, background noise, and environmental fluctuations. Along these lines, we did not predetermine specific SP-biological response relationships, not excluding the possibility of future discoveries. Overall, this agnostic approach was intended to lead to unbiased results.

The identified SPs were integrated using logistic regression to generate a score, where positive values indicated predicted susceptibility and negative values predicted resistance (Supplementary Information). While these scores represented numerical values, they were meant to be interpreted qualitatively as either susceptible (S) or resistant (R). It is important to understand that these scores did not directly correlate with MICs, and should not be compared across models using different SPs. All models achieved Pareto optimality[37], i.e., maximizing accuracy as the first criterion while minimizing the number of SPs as the second criterion. In other words, neither models with more nor fewer SPs performed better. The validation of these models occurred dynamically in a 3-fold cross-validation procedure repeated 300 times.

The improvement of the classification performance, depending on the number of SPs, is illustrated for the combination of *K. pneumoniae* and CIP. We evaluated 83 strains across 233 recordings. These strains presented a MIC distribution ranging from very susceptible (low MIC values) to very resistant (high MIC values) (Fig. 3a). A classification model built based on one SP delineated resistant and susceptible strains with an accuracy of 85.8%. This value increased up to 93.1% when subsequently a second, third, and fourth SP was added to the previous SPs at which point no further improvement of the performance was observed (Fig. 3b–d, Supplementary Data 2 and Supplementary Information).

### First general nanomotion AST for CRO and CIP

Similarly, we trained multi-SP-based classification models for *E. coli/K. pneumoniae* + CRO (Fig. 4a), *E. coli* + CIP (Fig. 4b), and *E. coli* + CTX (Supplementary Fig. 7) and used models with the number of SPs for which the performance saturated (Supplementary Information). In this setup, each sample from one blood pellet was measured in one or more technical replicates (=recordings). Multiple samples were obtained from some clinical isolates. For instance, a single isolate could be used for spiking an anaerobic and an aerobic blood culture – both used in hospitals. When multiple nanomotion recordings were obtained from the same sample, we obtained a score for each recording of a PBC sample and used the median score for sample classification and reporting of the final result.

lactamases (Supplementary Fig. 6). However, the available data for these strains are insufficient to exclude other influencing factors beyond resistance mechanisms.

### Classification algorithm development using machine learning based on SPs

To enhance the clinical applicability of the nanomotion AST, amid strain diversity, we needed multidimensional signal analysis techniques. Informative SPs were derived from power spectral density[45] (PSD), quantile spectrum[46], and multi-fractal detrended analysis[47] (MF-DFA), each providing unique insights into the signal's frequency- and time-domain properties (Supplementary Information, Methods). Employing ML algorithms instead of handpicking SPs, we extracted tens of thousands of SPs from 4-h nanomotion recordings across sets of strains and antibiotic combinations. These strains ranged from 83 strains of *K. pneumoniae* + CIP to 160 for *E. coli/K. pneumoniae* + CRO

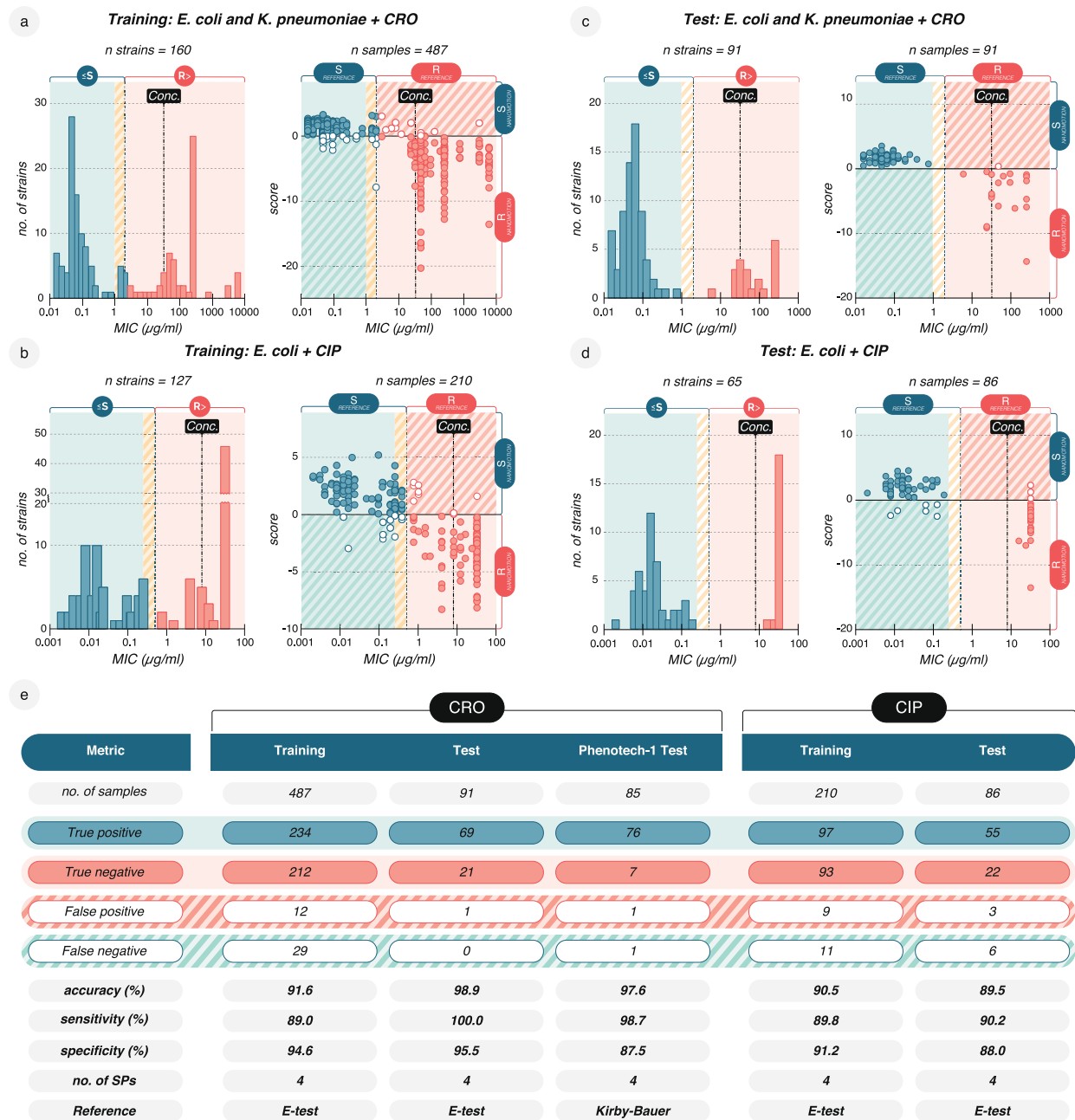

**Fig. 4 | Multiple-SP model nanomotion AST performance relative to MIC strips.**
**a** Left: To develop a model for CRO, the MIC distribution of 160 *E. coli* and *K. pneumoniae* isolates that were analysed in 4-h nanomotion recordings at 32 µg/ml CRO (*Conc.*, dashed line). Right: Classification according to nanomotion and a 4-SP model, with MIC strip serving as a reference. Circles represent median scores for a PBC samples measured ≥1 time (487 PBC samples and 1485 recordings, Supplementary Data 2). Score is defined as S > 0 > R. Closed circles show correctly classified measurements (TP, TN), open circles show falsely classified measurements (FP, FN). **b** Left: MIC distribution of 127 *E. coli* strains at 8 µg/ml CIP (conc. dashed line) analysed to develop a CIP model (210 PBC samples, 573 recordings). Right:

Classification according to nanomotion and a 4-SP model, with MIC strip serving as a reference. **c** Independent testing of the CRO model in (**a**) on 91 *E. coli* and *K. pneumoniae* isolates (left, MIC distribution) and classification (right), with MIC strip serving as a reference. **d** Independent testing of the CIP model in (**b**) on 65 *E. coli* strains (left, MIC distribution) and classification (right), with MIC strip serving as a reference. **e** Training and test performance for each of the two classification models including results from the PHENOTECH-1 study reporting accuracy [(TP + TN)/n], sensitivity [TP/(TP + FN)], and specificity [TN/((TN + FP)], with Kirby-Bauer serving as a reference. Data are available in the source data file. Single SP values and scores are available in Supplementary Data 2.

Early during development, we discovered that specific models utilizing different SPs for each antibiotic-species combination outperformed a general model. This is likely due to the nanomotion data's ability to capture the distinct mechanisms of action and reaction kinetics associated with each combination, even when comparing 3rd-generation cephalosporins like CTX and CRO.

The CRO model achieved 91.6% accuracy and relied exclusively on four spectral - the most intuitive -SPs. SP1 has the biggest impact on the

performance and describes the ratio between two time intervals: the integral of the PSD in the 20–28 Hz frequency range of 90–120 min and 0–30 min of the drug phase. We measured a balanced set of 160 susceptible and resistant strains across 487 spiked PBC samples, with 30% of the experiments using the *E. coli* reference strains ATCC-25922 and BAA-2452. The model was based on 1485 recordings (Supplementary Data 2). Similarly, the model for CIP was based on 127 strains across 210 PBC samples, achieving an accuracy of 90.5% using quantile and MF-

DFA SPs. To further validate the models' generalizability, we tested them on independent data sets comprising either spiked PBCs or direct isolates from anonymized patient PBCs. Importantly, none of the isolates used for training were reused for testing and neither feature selection nor classifier training was repeated on the new isolates. The results confirmed the models' applicability for a highly performing AST (accuracy$_{CRO}$ = 98.9%, accuracy$_{CIP}$ = 89.5%, Fig. 4c–e). The CRO model is being clinically validated on patient samples in the PHENOTECH-1 clinical performance study (NCT05613322). On a current sample size of 85 strains, including *E. coli* and *K. pneumoniae*, the model achieved an accuracy of 97.6% with a mean TTR of 4.24 h (SD = 0.21 h) across three different hospital sites (Fig. 4e). Here, results were again reported based on triplicate measurements.

Susceptible or resistant isolates further off the clinical breakpoints did not pose a problem for any of the classification models (Fig. 4a–d). However, samples closer to the breakpoints, including those classified as "susceptible with increased exposure" (EUCAST "I" clinical category), were more prone to incorrect classification, as seen with other phenotypic AST methods. This issue was most pronounced for CIP, as bacterial point mutations often confer low-level resistance to the antibiotic that only leads to a high MIC after further mutations accumulation[48]. Measurements of the low-resistant isolates NARA-5033, IHMA-2195830, and RYC-130 from different PBCs were repeatedly misclassified in the training (Supplementary Data 2). In the future, these models require training with more isolates exhibiting MICs around the breakpoints to better capture the nanomotion phenotype corresponding to the underlying low-resistance mechanisms.

### 4-h TTR nanomotion AST for antibiotic-inhibitor combination ceftazidime-avibactam (CZA)

The AST involved three *E. coli* isolates incubated with different concentrations of the last-resort cephalosporin/β-lactamase inhibitor combination ceftazidime-avibactam, considered strongly bactericidal. BAA-2452 and IHMA-2155385 exhibited resistance to ceftazidime alone, but IHMA-2155385 became susceptible to ceftazidime after the inhibitor avibactam was added, thereby inhibiting the extended-spectrum beta-lactamase (Fig. 5a). BAA-2452 remained resistant as it is known to produce a metallo-beta-lactamase (*blaNDM*). Similar to CRO, the SP slope decreased with rising ceftazidime concentrations at a consistent inhibitory concentration of avibactam (Fig. 5b). Using a balanced set of susceptible and resistant isolates (Fig. 5c, Supplementary Data 1) and our machine-learning pipeline, we developed a classification model that relies on a single SP (Supplementary Information). This model effectively differentiated susceptible isolates from resistant isolates, achieving an accuracy of 100% on recording level (Fig. 5d). The perfect performance is most likely attributed to the small dataset size and the limited diversity of resistance mechanisms towards ceftazidime-avibactam as a relatively newly deployed antibiotic compared to CRO or CIP.

### 2-h TTR nanomotion AST for CZA

We challenged the performance of the CZA 4-h AST by decreasing the recording time to 2 h (0.5 h medium phase, 1.5 h drug phase, Supplementary Fig. 8). This became feasible by updating the device with temperature control, allowing measurements at 37 °C to which human pathogens are well adapted and show quicker reactions. We increased the diversity of isolates by training the model on 25 *E. coli* and 21 *K. pneumoniae* isolates, anticipating a higher degree of diversity would necessitate a more complex model based on more information, i.e., ≥1 SP. Indeed, with 6 SPs, the training performance saturated at an accuracy of 95.8% (Fig. 6a, c, Supplementary Information). Following this, the 6-SP model was tested on an independent set comprising 17 *E. coli* and 4 *K. pneumoniae* isolates and demonstrated an accuracy of 93.0%. (Fig. 6b, c). This result supports that information in a 2-h

nanomotion recording is sufficient to build general classification models for a rapid TTR, further expediting the reception of actionable results in the clinic.

## Discussion

The nanomotion technology platform presented herein represents different approach to AST—one that does not rely on assessing bacterial growth. Our platform, by directly processing PBC samples and assessing bacterial cell vibration measurements, generated results in two (at 37 °C) or 4 h (at RT) instead of the 24 h required by current AST methodologies. Moreover, this method bypasses the plating step that usually occurs after PBC sample collection and before cartridge inoculation in current automated AST systems although polymicrobial samples were not tested in this study and could affect the results. During the training of classification models, this platform consistently achieved accuracy rates ranging from 90.5% to 100% for fluoroquinolones, cephalosporins, and cephalosporin-inhibitor combinations, demonstrating comparability to standard clinical diagnostic methods. Independent testing of these models further confirmed their generalizability, with accuracies ranging from 89.5% to 98.9%.

As of 2020, *E. coli* resistance frequency to CTX, CRO, and CIP was greater than 50% in some European countries[49,50]. To continue using these antibiotics rather than further increasing reliance on last-resort broad-spectrum agents such as carbapenems or cephalosporin/carbapenem-inhibitor combinations, clinicians need access to AST information to avoid inadequate antibiotic therapy. A drastic reduction in AST TTR is necessary to limit the time of empirical drug administration and early switch to an informed decision-based paradigm. With the set-up outlined in this study, a nanomotion-based AST takes 2 or 4 h to generate a sufficient recording starting from a PBC.

The classification models highlighted in this study were developed using clinical isolates from several hospitals and strain collections. Since these models are based on relatively few SPs, processing nanomotion signals and establishing a classification as either resistant or susceptible is a relatively quick process of a few minutes. The fact that the test does not rely on growth-based parameters, the fixed recording duration, and the automated data processing procedure ensure a standardized TTR that varies by only a few minutes from experiment to experiment. Indeed, this AST strategy is currently being tested for bacteremia and sepsis patients in two clinical studies (NANO-RAST[51] and PHENOTECH-1). The results reported here are those concerning the performance and TTR for one-third of the study population of the multicentric PHENOTECH-1 study.

Moreover, measurement parallelization is needed to address hospital sampling frequency requirements. Parallelization would also permit the simultaneous measurement of different drug concentrations, which is potentially required for agents like CIP, for which our method encountered difficulties around MIC-based clinical breakpoints. Moreover, parallelization enables rapid test results for multiple antibiotics, crucial for patients allergic to a tested drug or with documented resistance to first-line treatment.

This study focused specifically on two drug classes: cephalosporins and fluoroquinolones. Both are heavily used in the clinic and are susceptible to an increasingly diverse range of resistance mechanisms, leading to steadily rising resistance rates. This study is the first published large-scope dataset to investigate strain antibiotic resistance diversity with nanomotion technology. The novel analysis tools and classification modeling employed herein have enabled the utilization of complex information for AST development (e.g., signal acquisition at 60 kHz, advanced signal processing, computational power, and machine learning to identify and filter useful information). Given this, we believe a comprehensive nanomotion database containing information on sequenced strains will allow us to link an SP or group of SPs to a specific resistance mechanism or process in a bacterial cell. This

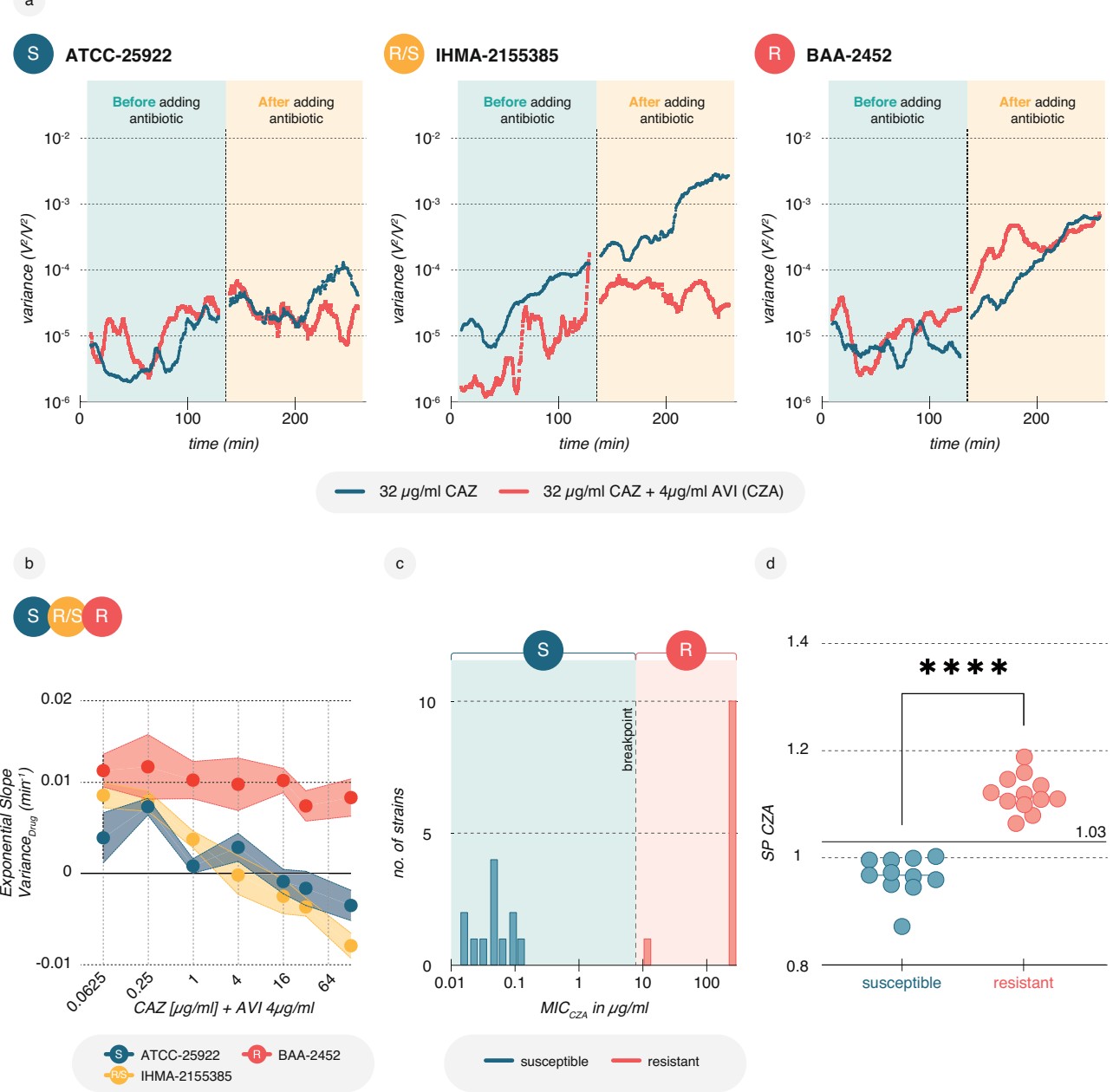

**Fig. 5 | Nanomotion AST for strongly bactericidal CZA based on 1-SP classification. a** Nanomotion recordings for three *E. coli* isolates: the susceptible (S) ATCC-25922 (left), the resistant to ceftazidime, CAZ, but susceptible to ceftazidime-avibactam, CZA, (R/S) IHMA-2155385 (middle), and the resistant (R) BAA-2452 (right). Strains were exposed to 32 μg/ml of CAZ either alone (blue) or in combination with 4 μg/ml of the β-lactamase inhibitor avibactam, AVI (red). The median variance of three recordings is shown. **b** SP "slope" for all three strains exposed a constant concentration of 4 μg/ml avibactam (AVI) as well as increasing concentrations of ceftazidime (CAZ). Each data point depicts the median of three experiments, with the shaded area marking the standard error. **c** MIC distribution of *E. coli* strains used to develop a CZA classification model. **d** A single-SP model perfectly discriminates all susceptible (S) and resistant (R) *E. coli* strains; nanomotion AST was performed on cells treated with a combination of ceftazidime (32 μg/ml) and avibactam (4 μg/ml). A two-tailed Mann–Whitney *U* test was used for statistical analysis, *p* ≤ 0.0001. Data are available in the source data file. Single SP values and scores are available in Supplementary Data 2.

would surpass the limitations of this study, where only the overall resistance phenotype could be identified.

Because it is growth-independent, nanomotion technology offers major advantages for assessing the susceptibility of slow-growing bacteria such as *M. tuberculosis*[34,36,52] or non-growing phenotypes found in many pathogenic bacteria[53–55]. Therefore, antibiotic tolerance, a state enabling bacteria to endure inhibitory concentrations without growing, can be measured. This method also assesses phage susceptibility[56] and evaluates drug effects on fungal[57] or cancer cells[58]. Therefore, nanomotion-based technology platforms such as Phenotech open the door for developing similar growth-independent rapid susceptibility testing platforms for antifungal and anticancer therapeutics.

## Methods

All experimental procedures and bioinformatic analyses in this work comply with ethical regulations and good scientific practices. An ethics approval for the pre-clinical experiments was not required as anonymized biological material, i.e., anonymized blood for the blood culture incubation, was provided by a blood donation center in

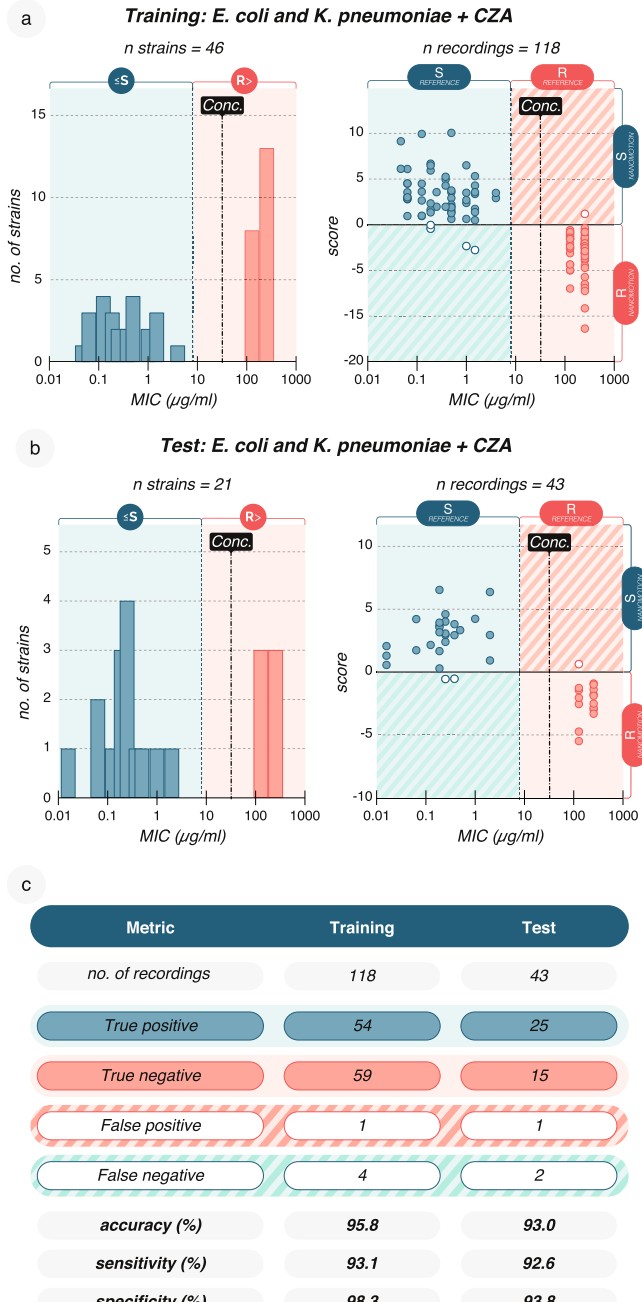

**Fig. 6 | Reduction of TTR to 2 h retains high performance for CZA. a** To develop a model for CZA on 2 h nanomotion recordings, the MIC distribution (*left*) of 46 *E. coli* and *K. pneumoniae* strains was analysed at 32 µg/ml CAZ (conc. Dashed line) and 4 µg/ml AVI. Right: Classification according to nanomotion and a 6-SP model, with MIC strip serving as a reference. **b** Independent testing of the 6-SP model in (**a**) on 21 different *E. coli* and *K. pneumoniae* isolates, *left*, MIC distribution, *right*, scores for each recording. **c** Training and testing performance of the 2-h model for CZA, accuracy [(TP + TN)/n], sensitivity [TP/(TP + FN)], and specificity [TN/((TN + FP)]. Data are available in the source data file. Single SP values and scores are available in Supplementary Data 2.

Switzerland. The clinical study protocol for the PHENOTECH-1 study (NCT05613322) was approved by the Ethics Committee for Investigation with Medicinal Products (CEIm) in Madrid (ID 239/22), the Cantonal Commission for Ethics in Research on Human Beings (CER-VD) in Lausanne (ID 2022-02085), and the Ethics Committee of the Medical University of Innsbruck in Innsbruck (ID 1271/2022).

## Bacterial strains and medium compositions

The strain collection used in this study consists of ATCC reference strains and clinical isolates either from patient blood samples at hospital sites or procured from strain collections (Supplementary Data 1). In order to establish a methodology for nanomotion-based AST, we used the *E. coli* reference strain ATCC-25922, which is susceptible to ceftriaxone (CRO; ceftriaxone disodium salt hemi(heptahydrate) analytical standard, Merck & Cie, Schaffhausen, Switzerland), cefotaxime (CTX; cefotaxime sodium, Pharmaceutical Secondary Standard, Supelco, Merck & Cie, Schaffhausen, Switzerland), ciprofloxacin (CIP; ciprofloxacin, VETRANAL®, analytical standard, Merck & Cie, Schaffhausen, Switzerland), and ceftazidime-avibactam (SigmaAldrich, Merck & Cie, Schaffhausen, Switzerland). Our reference strains for antibiotic resistance were BAA-2452 (resistant to CRO and CTX, *bla*NDM producer) and BAA-2469 (resistant to CIP). The *K. pneumoniae* reference isolates ATCC-27736 was susceptible to CRO.

To differentiate between resistant and susceptible phenotypes, clinical isolates were selected based on their MIC in accordance with the European Committee on Antimicrobial Susceptibility Testing (EUCAST) interpretation guidelines[59]. MIC strips and disk diffusion tests were performed on MH Agar plates (Mueller-Hinton agar VWR International GmbH, Dietikon, Switzerland). During all nanomotion experiments, bacteria in the measurement chamber were incubated with filtered (0.02 µm Polyethersulfone, PES, Corning, or Millipore) LB (Miller's LB Broth, Corning®) half-diluted in deionized water (Molecular Biology Grade Water, Cytiva), hereafter referred to as 50% LB.

All bacterial strains were stored at −80 °C in 20% glycerol. Bacterial samples for nanomotion experiments were prepared by first thawing new cell aliquots and growing them at 37 °C on Columbia agar medium solid plates (Columbia blood Agar, 5% sheep blood, VWR International GmbH, Dietikon, Switzerland). These cells were then used to inoculate blood culture medium and subsequently grown for nanomotion experimentation.

## Determining reference MICs (MIC gradient tests)

We performed MIC gradient tests (MIC strips) to determine the minimal inhibitory concentration (MIC) for each antibiotic used in this study. Cell suspensions were prepared by selecting three to five colonies grown overnight (ON) at 37 °C on a Columbia agar plate and resuspending them in 0.9% NaCl solution (Sodium Chloride, 0.9%, HuberLab, PanReac Applichem) at a density of 0.5 McFarland units (corresponding to $OD_{600nm}$ = 0.07). This suspension was then spread on MH plates using a sterile cotton swab to create a confluent culture. MIC strips (ceftriaxone 0.016–256 µg/mL, ciprofloxacin 0.002–32 µg/mL, cefotaxime 0.016–256 µg/mL, ceftazidime 0.016–256 µg/mL, and ceftazidime-avibactam 0.016/4–256/4 µg/mL MIC test strips, Liofilchem, Roseto degli Abruzzi, Teramo, Italy) were then placed onto inoculated plates using tweezers. The plates were subsequently incubated at 37 °C for 16–20 h, with the growth inhibition area surrounding the MIC strip present after this incubation period used to interpret MICs.

While MIC strips served as the primary AST reference method, some situations presented difficult interpretations or exceeded the scale of the CRO MIC strips. Here, broth microdilution assays were performed according to EUCAST recommendations[59]. Furthermore, a disk diffusion assay (DDA) was performed in parallel to each sample assessed using nanomotion technology for quality assurance purposes[20,60].

## Cantilever functionalization

To facilitate bacterial attachment and prevent cellular detachment during AST recording, we incubated the cantilever with 50 µl of 0.1 mg/ml PDL (Poly-D-Lysine hydrobromide, MP Biomedicals, Santa Ana, California, USA) diluted in molecular biology grade water

(HyClone, Logan, Utah, United States) for 20 min at room temperature (RT). This treatment created a homogenous positive electric charge that enabled the attachment of negatively charged bacteria. Following incubation, the PDL drop was removed and discarded, after which the cantilever tip was gently washed with 100 μl of molecular biology-grade water. The sensors on the cantilever were then allowed to dry for at least 15 min before use.

## Generating spiked bacterial blood cultures

Spiking refers to the process of inoculating blood culture samples with artificially infected blood. Here, we cultured strains of interest on Columbia Agar plates ON at 37 °C, isolated a single colony, and resuspended it in 0.9% NaCl with volumes adjusted to obtain a 0.5 McFarland density. We then performed two 1:10 serial dilutions, starting with that suspension, to generate a final dilution of 1:100. Finally, 10 μl of the final dilution were added to 9990 μl of EDTA blood from a donor provided by a blood donation center in Switzerland. Blood has been received fully anonymized.

To generate spiked blood cultures, we added 10 ml of artificially infected blood to either anaerobic (ANH) or aerobic (AEH) blood culture bottles (BD BACTEC™ Lytic Anaerobic medium and BD BACTEC™ Standard Aerobic medium Culture Vials; Becton Dickinson, Eysins, Switzerland) using a syringe. These culture bottles were then incubated until positivity, as determined by the BACTEC™ 9240 automated blood culture system (Becton Dickinson), was reached. In most cases, this process took 12 h or an overnight incubation.

## Preparing bacterial pellets from positive blood cultures

To generate and purify bacterial pellets for nanomotion recordings, we used either the MBT Sepsityper® IVD Kit (Bruker) or the direct attachment method (DA). When using the MBT Sepsityper® IVD Kit, we followed the manufacturer's instructions. Briefly, 1 ml of blood culture was combined with 200 μl Lysis Buffer, mixed by vortexing, and then centrifuged for 2 min at $12,000 \times g$ to obtain a bacterial pellet. The supernatant was discarded, while the bacterial pellet was resuspended in 1 ml of Washing Buffer. The resuspension was then centrifuged again for 1 min at $12,000 \times g$ to remove debris. For DA, 1 ml of positive blood culture (PBC) was syringe filtered (5 μm pore size, Acrodisc® Syringe Filters with Supor® Membrane, Pall, Fribourg, Switzerland). The pellet was then used for attachment to the cantilever.

## Attaching bacteria to the cantilever

Bacterial cells from prepared pellets needed to be immobilized onto the surface of the functionalized cantilever for nanomotion recording. First, pellets were resuspended in a PBS (Phosphate Buffer Saline, Corning) solution containing 0.04% agarose. Next, the sensor was placed on a clean layer of Parafilm® M (Amcor, Victoria, Australia). The tip of the sensor, containing the chip with the cantilever, was placed into contact with a single drop of bacterial cell suspension for 1 min. After this, the sensor was removed, gently washed with PBS, and assessed using phase microscopy for attachment quality. In the event of unsatisfactory attachment, the sensor was re-incubated in the cell suspension for an additional 30–60 s, or until satisfactory attachment was achieved. We aimed for an even bacterial distribution across the sensor (Fig. 1b, c, and Supplementary Fig. 2). The attachment of bacteria is part of a filed patent (PCT/EP2020/087821).

## Nanomotion measurement platform

Our nanomotion measurement platform, the Resistell Phenotech device (Resistell AG, Muttenz, Switzerland), comprises a stainless-steel head with a measurement fluid chamber, an active vibration damping system, acquisition and control electronics, and a computer terminal.

Nanomotion-based AST strategies utilize technologies that are well-established in atomic-force microscopy (AFM). Specifically, our nanomotion detection system is based on an AFM setup for cantilever-based optical deflection detection. However, in contrast to standard AFM devices, in the Phenotech device the light source and the photodetector are placed below the cantilever to facilitate the experimental workflow. A light beam, focused at the cantilever end, originates in a superluminescent diode (SLED) module (wavelength: 650 mm, optical power: 2 mW), is reflected, and reaches a four-sectional position-sensitive photodetector that is a part of a custom-made precision preamplifier (Resistell AG). The flexural deflection of the cantilever is transformed into an electrical signal, which is further processed by a custom-made dedicated electronic module (Resistell AG) and recorded using a data acquisition card (USB-6212; National Instruments, Austin, TX, USA). The device is controlled using a dedicated AST software (custom-made, Resistell AG).

The custom-made sensors used for the described experiments (Resistell AG) contain quartz-like tipless cantilevers with a gold coating acting as a mirror for the light beam (SD-qp-CONT-TL, spring constant: 0.1 N/m, length × width × thickness: $130 \times 40 \times 0.75$ μm, resonant frequency in air: 32 kHz; NanoWorld AG, Neuchâtel, Switzerland). During an AST experiment, bacterial nanoscale movements actuate the cantilever to deflect in specific frequencies and amplitudes.

For the development of temperature-controlled experiments with CZA at 37 °C, we used modular NanoMotion Device (NMD) prototypes. It allowed the reconfiguration of the hardware setup to work with either a standard incubator or a modified measurement head to warm up only the measurement chamber. For the merge of an NMD with a BINDER BD 56 incubator, the size of the incubator fits the entire NMD head with the active vibration damping module, also permitting the user a comfortable manual operation. The incubator shelf was rigid and able to hold the vibration isolator and NMD head (ca. 10 kg), and the incubator was modified with an access port to pass through control cables operating the light source, photodetector, and vibration damping module from the outside. Another NMD prototype was equipped with a locally-heated measurement chamber, thermally insulated from the measurement head set-up. A Peltier module as a heating element was installed under the measurement chamber, adapted to temperature control by adding a Pt100 temperature sensor. Temperature was kept at 37 °C by a Eurotherm EPC3016 PID controller (Eurotherm Ltd, Worthing, United Kingdom) and a custom-made Peltier module driver. Both setups had a temperature stability <0.2 °C, which is a matching requirement for stable culture conditions.

## Calculating variance over time and slope analysis

Each sampled nanomotion signal was split into 10 s timeframes. For each timeframe, the linear trend was removed and the variance of the residue frame was estimated. For some experiments, the variance signal was too noisy for classification, necessitating the application of an additional smoothing procedure. A running median with a 1 min time window was applied to smooth the variance signal and allow plot interpretation. For the calculation of the SP slope of the variance in the drug phase used for determining the nanomotion dose response in Fig. 2b and Supplementary Fig. 4, we used the formula $\log(x) = \log(C) + at$, where t is time (in min), a is the slope of the common logarithm of the variance trend, and $\log(C)$ is the intercept. Variance plots were used here for the visual inspection of results, and are currently the primary tool accessible for investigators. However, more sophisticated SPs are necessary for reliably classifying phenotypes in ASTs.

## The Resistell nanomotion-based AST experimental setup

Nanomotion-based AST was performed using Resistell Phenotech devices (Resistell AG, Muttenz, Switzerland) on a standard laboratory benchtop. Each recording comprises two phases: a 2-h medium phase and a 2-h drug phase. In addition, a short blank phase is conducted to measure the baseline deflections of a new, bare, functionalized cantilever in 50% LB medium for 5–10 min. Raw nanomotion recordings were used to develop classification models using machine learning.

The signal during the blank phase is expected to be constant and primarily flat (variance around 2.6 E-6 or lower). Higher median values or the clear presence of peaks are indicators of potential contamination of the culture medium inside the measurement fluid chamber, sensor manufacturing errors, or an unusual external environmental noise source that should be identified and rectified. In particular, contamination ($OD_{600} < 0.01$) can cause deflection signals that are several orders of magnitude higher than expected for sterile media due to interactions between floating particles in the fluid chamber and the laser beam. The blank phase serves as a quality control but is not used for classification models and, therefore, can be performed several hours prior to recording medium and drug phases.

The medium phase records cantilever deflections after bacterial attachment, showing the oscillations caused by natural bacterial nanomotions stemming from metabolic and cellular activity. Here, variance is expected to be greater ($10^{-5}$ to $10^{-3}$) than during the blank phase. The 2-h medium phase duration allows cells to adapt to their new environment within the fluid chamber and generates a baseline that can be compared to bacterial nanomotions during the drug phase. The drug phase measures cellular vibrations after an antibiotic has been introduced to the fluid chamber. The antibiotic is directly pipetted into the medium already present within the measurement chamber.

## Machine learning and development of classification models

The Phenotech device detects nanomotion signals resulting from the activity of living cells. However, other sources can create detectable noise during cantilever-based sensing[61]. Thermal drift occurring on the cantilever[62], as well as external sources such as acoustic noise and mechanical vibrations, can all impact measurements. Distinguishing cell-generated vibrations from background noise can be challenging. As such, we employed a supervised machine learning-based approach to extract signal parameters (SPs) containing diagnostic information while minimizing overall background noise. The entire procedure of analyzing motional activity of particles is part of a filed patent (PCT/EP 2023/055596).

First, a batch of initial SPs related to frequency and time domains were extracted, with time and frequency resolution being high to allow for further statistical analysis at this level. Next, different statistical parameters were created with a much coarser time and frequency scale. Finally, various combinations (differences, ratios, etc.) were calculated, forming a final batch of SPs that are more related to antibiotic susceptibility. SPs were estimated for experiments with cells and conditions with well-defined and known outputs (e.g., susceptibility to a given antibiotic could be known through reference AST methods). Here, extracted SPs and outputs formed labeled datasets that could be used for supervised machine learning.

A feature selection algorithm extracted SPs related to the phenomenon of interest. These SPs were selected from the overall batch of SPs to optimize the performance of this so-called machine learning model. In this case, the model was a classifier validated by analysis of metrics measuring the degree of distinguishing antibiotic susceptibility. Therefore, a forward selection method was applied. All SPs were subsequently evaluated in the classifier with repeated stratified cross-validation. The SPs that enabled the classifier to reach maximal accuracy were added to the stack of selected SPs and deleted from the remaining SPs. In the next iteration, all remaining SPs were again tested with the already-selected SPs. The best-performing SP was again added to the selected SP stack. This process was repeated several times until the overall performance reached a plateau or a predefined number of SPs were selected. In the final model (iii), these newly found SPs were then used as machine learning model features. Classifier models were trained using the complete available dataset and could now be used to classify previously unseen data. The Supplementary information elaborates in more detail on that process and lists all SPs used in the different classification models.

## Independent testing of classification models

After achieving Pareto optimality, the models were tested on independent test datasets consisting exclusively of strains of *K. pneumoniae* or *E. coli* that were not used in the training of the corresponding model. We used either spiked blood cultures or directly anonymized remnant PBC from the Lausanne University Hospital (CHUV) in Lausanne. Spiking was predominantly utilized to increase the fraction of resistant strains to obtain more representative specificity (classification performance of resistant strains), as resistance rates at that hospital are around 10 % for CRO and CIP and close to non-existent for CZA. Each nanomotion recording was classified separately and combined using the median to a sample reporting accuracy, sensitivity and specificity – exactly as described for reporting the training performance.

In addition to this, we performed an interim analysis of the multicentric clinical performance study PHENOTECH-1 (NCT05613322), conducted in Switzerland (Lausanne University Hospital, Lausanne), Spain (University Hospital Ramón y Cajal, Madrid) and Austria (Medical University of Innsbruck, Innsbruck). The study evaluates the performance of the nanomotion AST with the Phenotech device using the CRO model on *E. coli* and *K. pneumoniae* from fresh residual PBC. Ethical review and approval were obtained by the hospital ethics committee at each participating site. In Lausanne and Innsbruck, only samples from patients who had previously agreed to the use of their residual biological material were utilized. In Madrid, consent for participation was not required for this study in accordance with institutional requirements. No compensation was paid to participants. The interim results reported here comprise the first included 85 samples with complete data entry. The eventual sample size of 250 was estimated based on the expected rate of *E. coli* and *K. pneumoniae* samples susceptible to the antibiotic in the three countries (i.e., 80%). Allowing for up to 10% samples with missing data or technical errors, an overall sample size of 250 would include 180 truly susceptible samples with 98% power to demonstrate that sensitivity is at least 90%. The PHENOTECH-1 study is expected to conclude in 2024. The endpoints of this study include the accuracy, sensitivity, and specificity of the device according to ISO-20776-2 (2021), as well as the time to result from the start of the AST to the generation of the result in form of a time stamped report. Regarding inclusion criteria, patients aged 18 years or older, with positive blood cultures for either *E. coli* or *K. pneumoniae*, are eligible for participation in the study. Additionally, Phenotech AST needs to be performed within 24 h of the blood culture turning positive. Patients with polymicrobial samples are excluded from the study.

Qualitative results of the Kirby Bauer disk diffusion assay, i.e., either R or S, were used for benchmarking. Clinical breakpoints for the class definition were according to EUCAST in 2022. The samples coming from one PBC were measured in technical triplicates for 4 h. The results from each recording were automatically combined to a sample. Instead of the median score, a majority voting system was in place that is, RRR, RRS and RR- return predicted resistance, SSS, SSR, SS- return susceptibility. In this way even if one recording needed to be excluded because of technical errors, or detection of substantial elongation of the specimen, the sample could be interpreted. Only if two or more recordings were excluded, or the exclusion of one recording resulted in the disagreement between the two remaining recordings, the sample would be classified as non-conclusive. The experiments were not randomized and the investigators were unblinded during experiments and outcome assessment. Information on sex, gender, and age of participants was not collected in this study as having no impact on the generalizability and translation of the

findings. At the time of analysis, the data set included 119 samples, of which 12 screening failures, 5 with technical errors or elongation, and 12 incomplete/unverified. Samples with complete, verified and cleaned data accounted to 90. Of these, the first 85 samples were selected of which 20 samples derived from CHUV, 48 samples from Ramon y Cajal Hospital and 17 samples from Medical University of Innsbruck.

### Statistics and reproducibility

Statistical details can be found in the figure legends. Data are presented as mean or median ± SD or representative single experiments and provided in the Source data file. In Figs. 3, 5, and 6, the performance calculation is based on single recordings for which a score was calculated. Each recording is depicted as a datapoint representing a biological replicate originating from a different PBC. Performance calculation in Fig. 4 is based on the median of the scores calculated for each technical replicate originating from the same PBC. Thus, each datapoint represents the median score as it is currently implemented in the PHENOTECH-1 clinical performance study. In each case, scores are logits predicted by the corresponding logistic regression model. In Fig. 5e the two-tailed Mann–Whitney $U$ test was performed for calculating a p-value. Statistical analysis and graphs were generated with GraphPad Prism 10.

### Reporting summary

Further information on research design is available in the Nature Portfolio Reporting Summary linked to this article.

## Data availability

Data associated with figures is available in the Source data file and the Supplementary data files. A comprehensive list of bacterial strains is provided in Supplementary Data 1. For the development and testing of classification models, detailed information about recordings, their connection to samples, and associated strains can be found in Supplementary Data 2. Here, we also list all SP values and scores. The description of each SP used in these models as well as all Supplementary Figs. can be found in the Supplementary Information file. The clinical study protocol for Phenotech-1 is summarized at https://clinicaltrials.gov/study/NCT05613322. Source data are provided with this paper.

## Code availability

The code used in this study for the calculation of the scores is available under https://github.com/resistell-com/nat-commun-ast-ml.

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

## Acknowledgements

For sharing strains and clinical isolates, we thank Prof. Dirk Bumann at the Biozentrum Basel, Prof. Jan Gorm Lisby at Hvidovre Hospital in Copenhagen, Dr. Antony Croxatto at CHUV in Lausanne, Dr. Michael Oberle and Ms. Nadine Hunn from Kantonsspital Aarau, Prof. Patrice Nordmann at the Swiss National Reference Center for Emerging Antibiotic Resistance (NARA) and Dr. Stephen Hawser at the International Health Management Associates (IHMA). We are grateful to Mr. Piotr Grzywacz for graphical design. We thank Pfizer for its generous financial support of the ceftazidime-avibactam project. Resistell received funding from the European Union's Accelerator grant RAPID-SEP-AST 961141. Resistell and G.G. received funding from Innosuisse Swiss Agency for the Promotion of Innovation grant 36334.1 IP-LS. S.K. was supported by Swiss National Science Foundation (SNSF) grant CRSII5_173863.

## Author contributions

D.C., A.S., G.G., C.O. and G.J. were responsible for the conception and design of the project. M.P.V., L.M., A.V., A.L., K.F. and E.D. conducted experiments. G.J. and G.C. developed the classification algorithms. Data analysis was performed by G.J., A.S., M.P.V., L.M., G.C. and C.O.; G.W., M.S. and R.B. were responsible for the technical development of the Phenotech devices. The clinical study team consisted of R.C., G.G., C.L.F., R.G., M.G.C., A.D. and F.T. providing and analysing clinical data

for testing of the classification algorithms. C.O. manages the Phenotech-1 study. A.S., D.C., G.J., G.C., M.S., G.W., A.L., S.K., C.O. and G.G. wrote the manuscript. Supplementary Information and Supplementary Data 1 and 2 were compiled by A.S. and G.J.

## Competing interests

S.K., R.C., and G.G. are scientific advisors to Resistell. E.D., A.L., M.S., and G.W. filed for Resistell the patent PCT/EP2020/08782 concerning the bacterial attachment to a cantilever. G.J., A.S., E.D., and D.C. filed for Resistell PCT/EP 2023/055596 about the ML analysis of particle motions on a cantilever. Resistell holds exclusive license for EP2766722B1 patenting the nanoscale motion detector of which S.K. is listed as one of the inventors.
