## [Peer Review File · Nature Communications]

REVIEWER COMMENTS

Reviewer #1 (Remarks to the Author):

SUMMARY:

The authors describe a bacterial nanomotion detection platform that enables prediction of antibiotic susceptibility within four hours based on the changes in vibration of a bacterium after exposure to an antibiotic. While the use of nanomotion for AST classification has been previously described, in this study, the authors applied machine learning algorithms to identify those signal parameters (SPs) extracted from nanomotion tracings that are most predictive of specific resistance patterns to cephalosporins and fluoroquinolones. They used this approach on a sample set of about 350 *E. coli* and *K. pneumoniae* isolates and report between 90-100% accuracy via internal cross-validation on this set of isolates. While the topic and approach are of interest, concerns remain about the possibility of overtraining on these specific strains.

MAJOR CRITIQUES:

1. The major limitation of the study is in the risk of overfitting. The authors explore hundreds of thousands of SPs and assess their performance in hundreds of isolates. It appears they perform feature selection on the same set as they assess performance. Even with multiple rounds of blinded cross-validation (in this case, 3-fold cross-validation performed 300 times), the vast excess of parameters over samples to be classified is a setup for overfitting to parameters that idiosyncratically fit the training set but may not generalize beyond. Once feature selection is complete, the authors should test a completely independent validation set of strains, ideally of at least comparable size to the training set but fully independent (i.e. different isolates, not just different measurements of the same isolates), to assess classification accuracy using only the 1-5 SPs selected from the separate training set. It appears from the main article and the supplemental text that the same isolates were used for both feature selection and performance assessment. If this is not true, and the authors did test an independent set of strains after feature selection was complete, this should be explicitly clarified. Otherwise, such independent validation should be done, ideally with geographically and/or phylogenetically distinct isolates.
2. The authors discuss signal parameters throughout the paper (e.g., lines 88, 131, 146, 160, 216-218), with some details in the Supplementary Information about how these signal parameters are generated from extracted features of the nanomotion data. However, it is very difficult to understand what is actually being measured for each selected signal parameter; the examples listed in the Supplement are very difficult to interpret. More interpretable explanations for the users of what SPs were selected for each bug-drug pair would be helpful. If this is not possible because of complexity, and the intent is to view these SPs as a black box, this should be more explicitly stated as well. Assuming independent validation (see Major Critique 1), intuitively understandable SPs are not a requirement for a useful classifier, but especially at this early stage with limited testing, if SPs are explicable in some manner that relates to drug response, that might increase confidence in their generalizability. The attempt on lines 216-218 ("The SP in question for this model was related to the time dependence of the local minimum in the power spectrum estimated at the beginning and end of the drug phase") is quite confusing.

3. The use of a single antibiotic concentration so far from the clinical breakpoint, chosen from the responses of only two strains, to classify susceptibility is puzzling, particularly in light of the lack of a clear relationship between their SPs and MIC (lines 141-150). And while it seems they must have determined the optimal concentration for each bacteria-antibiotic combination, it is only discussed for ceftriaxone – a similar figure to Fig 2b, used to determine each dose for each pathogen, might be added for other pairs as a Supplemental Figure. Do the authors envision using a single dose for each antibiotic, for all pathogens? Or might pathogen identity impact the exposure dose? Independent concentration optimization on *K. pneumoniae* could help verify the approach, given the similarity between the two species.

4. Lines 195-202: not clear how helpful more training with isolates near the breakpoint MIC will actually be for classification accuracy, given the “non-intuitive” relationship between SPs and MIC (lines 141-150) – it seems that making fine distinctions around the breakpoint would require a predictable relationship between SPs and MIC. It is also not clear what the authors mean by the last sentence on line 205 in this context – what “additional information for fast clinical decision-making” would be added by inaccurate classification near the breakpoint MIC? Or is this referring simply to the accuracy for isolates farther from the breakpoint MIC?

MINOR COMMENTS:

1. Lines 163-165: the “score” referred to here, and plotted in Fig 3-4, is not clearly described in the Supplement. Given its use in main figures, a description of this score is warranted in the main text.

2. Lines 165-167: From a clinical perspective, lumping “susceptible with increased exposure” together with “susceptible” is fraught, since dosing would be different for the two states (and this is felt to be important for clinical outcomes), yet presumably this assay as currently configured would not make this distinction.

3. Lines 182-183: handling isolates tested multiple times does require some special consideration, but it is not clear that using the median SP value from multiple measurements is appropriate – if the test is not to be deployed on median measurements, then using them here for accuracy reporting seems questionable. Instead, perhaps one test per isolate could be selected at random for overall accuracy reporting, whereas the classification of multiple replicates of the same isolate could be separately tabulated as a measure of assay precision?

4. Line 188: For CRO testing, 30% of the experiments apparently used just two isolates (the reference *S & E coli*) – were these overrepresented in the training sets, or was only the median used? Was this imbalance true for all bacteria-antibiotic combinations?

5. Figure 3 and elsewhere: “false positive” and “false negative” for classification are unclear (does “positive” refer to resistant or susceptible?); “Major Errors” and “Very Major Errors” are more standard terms for AST.

Reviewer #2 (Remarks to the Author):

The article by Sturm et al. proposes the use of machine learning and nanomotion data acquired from an AFM cantilever for classifying resistant from susceptible bacteria. The measurements and data

processing are carried out on a library of clinical isolates of *E. coli* and *Klebsiella Pneumonia*, and suggest that the sensitivity analysis can be carried out using machine learning with good accuracy.

Although I admire the large amount of data analysis and experiments performed by the authors, unfortunately, the organization of the article, its structure and the findings does not have enough merit to justify its publication in *Nature Communications*. My reasonings are brought below.

1. Undoubtedly nanomotion technology has the potential to become an important player in the future of Antibiotic Susceptibility Testing (AST) market. The use of AFM cantilevers for this technology, as authors have used, is known for years and has been already highlighted in several review articles. Therefore, the experimental part of the article is not new and authors do not bring new advancements on that front. As a result, many of the statements in the introduction and conclusion sections related to nanomotion technology are already pretty well-known facts and neither challenges our current understanding nor pushes it to the next level. The machine learning part is also not new as authors seem to use classifying algorithms (though unclear what are these classifiers- which I reflect on later). Besides, the combination of both, nanomotion and machine learning, does not seem to be a new idea either. In fact, the same group published this idea in a similar article (see: *Microbes and Infection* (2023): 105151). Therefore, the present article does not seem to further advance our experimental understanding of nanomotion technology nor offers new insights.

2. Following up on the use of machine learning, it is unclear what type of classifier has been used and why. This is at least not clear from the main text, and the signal analysis part in the SI is prepared in such a sub-optimal fashion that it makes it difficult to follow. Only in several instances in the text authors mention that the models used were all Pareto optimal, while it remains ambiguous how did authors find this optimality? What hyper-parameters in what classification algorithms they were changing to find the Pareto optimal condition. As a result, the reader remains confused how this classification and analysis is performed.

3. Presumably, the only new element of the article is the use of, what authors call “signal parameters” (SP) instead of variance of a signal as a metric to perform sensitivity analysis. The authors rationale in coming up with these SPs is that the slope of the variance does not allow them to distinguish the sensitivity of phenotypes across different concentrations. Thus, they come up with new metrics/SPs. Unfortunately, these SPs are not even mentioned once in the whole manuscript, although they seem to be the main new thing the article is offering. The SPs are brought in a very disorganized fashion in the SI, making it difficult for the reader to identify at the end how many SPs and what has been used in authors classification. For instance, looking at Fig 3b, where authors show improvement in performance by adding more SPs, it is not clear what parameter is SP1, or what parameters authors considered for SP=3 or SP=4, and why for instance SP=5 is not there in the figure and what this SP class really entails?

4. Regarding the SP matter, it is also not clear how these SPs change before and after adding antibiotic. There is only one schematic in the SI that to some extent defines SPs – But variation of SPs for real data are completely absent, and thus their sensitivity could be a matter of question. For instance, the dip that can be seen in PSD has been suggested to serve as an SP. But, the system under investigation is pretty noisy, there is noise in amplitude and frequency, therefore, it is unclear how accurate such a SP could be used in performing the analysis. Besides, it could be that this SP is fully masked under the nanomotion noise and be strain dependent.

5. Following up on the same, for instance two frequencies around cantilever resonance are chosen as SPs. I doubt such SPs could be a good choice as they very much depend on the experimental conditions and the cantilever properties. It could be very well that there are slight changes in the stiffness/mass of the cantilevers in different experiments which could lead to different SPs and could easily influence the classification procedure. Even if we assume that the cantilevers in all experiments were closely identical, the initial cell density/mass is highly varying based on Extended Fig 2. This becomes even more important considering that the resonance peak and the frequencies nearby remain unaffected in the presence of antibiotics in nanomotion experiments: see for instance Fig. 3 in <http://dx.doi.org/10.1063/1.4895132>, which clearly supports points 4 and 5.

6. As a result, likely most of the SPs chosen by the authors are independent of the nanomotion signal itself and do not seem to have clear correlation with the biological fingerprints. Therefore, other SPs shall be used for classification purposes that are biologically relevant. One such SP, that apparently authors also considered (but again it is not clearly mentioned) is the slope of the spectrum. However, it is pretty well established that the PSD of *E. coli* is characterized by $1/f^a$ where a has been obtained independently by different groups to be close to 2, and that the spectrum becomes white if the bacteria are dead. Therefore, it is also not clear what new information such a metric/SP can offer other than what can already be entailed from variance.

Also, few more unclarities related to the experiments and sample preparation:

- The authors compare susceptibility data obtained by EUCAST conditions to experimental conditions but performed their own experiments in 50% LB medium + 50% deionized water, which is very different from EUCAST conditions. The growth rate and doubling times of microorganisms might be very different in their case versus EUCAST. In their earlier study (Nature Nano, 2013) the authors claim that the nanomotion couples with the growth rate of the microorganisms.
- Why is there such a high variation in the nanomotion variance (and often a clear upward trend) during the 2h measurement even without the addition of the antibiotic? (see eg. Fig1d and fig.2c , fig 5a, fig. s3).
- It is unclear why did the authors use a high concentration of the CRO antibiotic (32ug/ml) , considering the breakpoint is at 1ug/ml?
- What is the role of gelling agent, agarose, in the sample preparation? It is not mentioned in the article.
- Table 2 is missing even though it is mentioned in the text.

For all the above, I do not recommend this article for publication in Nature Communications.

Reviewer #3 (Remarks to the Author):

In the manuscript titled "Accurate and rapid antibiotic susceptibility assessment using a machine learning-guided nanomotion technology platform," the authors present an integrated nanomotion technology platform for measuring bacterial response to antibiotics. Nanomotion technology, based on atomic force microscopy, provides a growth-independent approach to assessing bacterial response. The platform combines hardware and software components, enabling fast and reliable analysis of large

datasets. Machine learning algorithms were employed to accurately classify antibiotic-resistant and -susceptible bacteria. The study focuses on testing the response of *E. coli* and *K. pneumoniae* strains to four commonly used antibiotics for bloodstream infections.

The nanomotion technology platform presented in the study offers a novel approach to antibiotic susceptibility testing that overcomes the reliance on bacterial growth assessment. As clearly described by the authors, it demonstrates faster results compared to existing methodologies, with a turnaround time of four hours instead of 24 hours. The platform eliminates the need for plating and enables a quicker overall process for obtaining patient antibiograms. Importantly, it achieves high classification accuracy rates (90.5% to 100%) for fluoroquinolones, cephalosporins, and cephalosporin-inhibitor combinations, comparable to standard clinical diagnostic methods.

In summary, the manuscript is well-written, with very clear figures, and I recommend its publication without any reservations. However, I have a few minor suggestions and questions for clarification that could enhance the manuscript's clarity and impact.

1. In the results session, regarding the nanomotion recordings:

a. It would be beneficial to provide more information on the specific resistance mechanisms that may be causing the observed increase in signal variance at sub-MIC concentrations of CRO. This clarification would help readers understand the underlying biological processes contributing to the nanomotion signals.

b. Please further explain the reasons behind the lack of correlation between the nanomotion signals and MIC values. It would be valuable to discuss potential factors or variables that could influence the nanomotion signals independently of the MIC values.

c. In order to ensure the generalizability of the nanomotion AST, it is important to address the diverse responses observed among different strains and antibiotics. Please elaborate on your plans to tackle this issue and provide details on any strategies you have in mind to account for the variability.

d. It would be helpful to provide more details on how the nanomotion signals correlate with bacterial vibrations rather than growth. This explanation would enhance the understanding of the physical phenomena underlying the nanomotion technology.

2. In the results session, regarding the classification algorithm:

a. Please provide more details on the specific machine learning algorithms used for the development of the classification models in the nanomotion AST. It would be beneficial to include this information in the main manuscript rather than solely in the Supplementary Information, as it will facilitate comprehension for readers.

b. How did you determine the optimal number of signal parameters (SPs) to extract from each nanomotion recording? Please elaborate on the methodology used to determine the appropriate number of SPs and any considerations taken into account during this process.

c. Can you explain the process of repeatedly selecting SPs during the development of the classification models? Please provide an overview of the methodology used and any criteria used to guide the selection process.

d. Please elaborate on the scoring system introduced for the classification models and how it relates to predicting susceptibility and resistance. A more detailed explanation of the scoring system will help readers understand its practical implications and its connection to the prediction outcomes.

3. In the results session, regarding the applicability of the methodology.

Please explain the rationale behind selecting only one signal parameter (SP) for the classification model to discriminate susceptible and resistant strains of *E. coli* treated with ceftazidime-avibactam. Providing an explanation for this specific choice will help readers understand the reasoning behind the decision and the potential advantages or limitations associated with using a single SP in this context.

Overall, the manuscript is well-written and provides a valuable contribution to the field of antibiotic susceptibility testing. Addressing these suggestions and providing further clarification will enhance the manuscript's clarity and impact.

**Reviewer #1 (Remarks to the Author):**

SUMMARY:

The authors describe a bacterial nanomotion detection platform that enables prediction of antibiotic
susceptibility within four hours based on the changes in vibration of a bacterium after exposure to an
antibiotic. While the use of nanomotion for AST classification has been previously described, in this study,
the authors applied machine learning algorithms to identify those signal parameters (SPs) extracted from
nanomotion tracings that are most predictive of specific resistance patterns to cephalosporins and
fluoroquinolones. They used this approach on a sample set of about 350 E. coli and K. pneumoniae isolates
and report between 90-100% accuracy via internal cross-validation on this set of isolates. While the topic
and approach are of interest, concerns remain about the possibility of overtraining on these specific strains.

MAJOR CRITIQUES:

1. The major limitation of the study is in the risk of overfitting. The authors explore hundreds of thousands
of SPs and assess their performance in hundreds of isolates. It appears they perform feature selection on the
same set as they assess performance. Even with multiple rounds of blinded cross-validation (in this case, 3-
16 fold cross-validation performed 300 times), the vast excess of parameters over samples to be classified is a
17 setup for overfitting to parameters that idiosyncratically fit the training set but may not generalize beyond.
Once feature selection is complete, the authors should test a completely independent validation set of
strains, ideally of at least comparable size to the training set but fully independent (i.e. different isolates,
not just different measurements of the same isolates), to assess classification accuracy using only the 1-5
SPs selected from the separate training set. It appears from the main article and the supplemental text that
the same isolates were used for both feature selection and performance assessment. If this is not true, and
the authors did test an independent set of strains after feature selection was complete, this should be
explicitly clarified. Otherwise, such independent validation should be done, ideally with geographically
and/or phylogenetically distinct isolates.

*We agree with the reviewer, the risk of overfitting exists. We included in the new version entirely*
*independent test sets (i.e., different experiments with different strains not used in the corresponding training*
*dataset). The original risk however was minimized by working with many different, representative isolates,*
*i.e. having an extensive training sample size, and keeping the number of signal parameters small. Therefore,*
*we saw the best training performance for Ec/Kp + CRO with 91.6 %. All models trained were cross-*
*validated before tested on the independent test set and none of the originally presented models were*
*retrained, i.e., one can exclude that information from the test sets were used to potentially improve the*
*original models.*

*As recommended by the reviewers, we added the following independent test sets*

*1) For Ec/Kp + CRO we tested the presented model on two independent test sets of strains not used in the*
*training (total n: 176), please also refer to the updated Suppl Table 2 to see the strains used for each*
*model training and testing:*

*a) A mixed test set consisting of anonymized blood culture samples from the University hospital in*
*Lausanne (CHUV) as well as spiked blood cultures from other sources to increase the number of*

resistant strains. The performance is summarized in Figure 4e (column title “CRO test”). The individual
 score for each sample vs MICs (MIC strip) is plotted in Figure 4c. It contains Ec and Kp. Sample size:
 91. Accuracy of the CRO model: 98.9%

b) A test set of positive blood cultures collected in the running clinical performance study PHENOTECH-
 1 (ClinicalTrials.gov ID: NCT05613322). The performance is summarized in Figure 4e (column title
 “CRO Phenotech-1”) with an accuracy of 97.6 %. It contains 85 Kp and Ec isolates from three different
 hospitals (University Hospital Ramon y Cajal Madrid, Medical University Innsbruck and University
 Hospital Lausanne). We do not collect MIC values in this study but compare it only qualitatively to
 Kirby- Bauer Tests (disk diffusions) without recording the diameter of the inhibition zone. These results
 were analyzed in the scope of the first interim-analysis (1/3 of samples included, i.e., 85 samples). The
 interim analysis report and study protocol are attached for the reviewers.

2) For Ec + CIP we tested the model on 86 Ec isolates. Some were anonymized blood cultures. The majority
 though were spiked blood culture samples again to increase the fraction of resistant isolates. The
 performance is summarized in Figure 4e (accuracy 89.5%) and the individual scores vs MIC is shown
 in Figure 4d.

3) We trained a new 2-hour model for CZA on Ec and KP and tested it on 21 different Ec/Kp strains whose
 performance is summarized in Figure 6 (spiked blood cultures). Training accuracy: 95.8 % and Testing
 accuracy 93.0 %.

4) We moved the Ec CTX model to the supplement as we were (i) unable to generate a sizable test set in
 the 3 months, (ii) it is partially redundant to CRO, (iii) we wanted to leave room for three test sets for
 CRO and CIP and the new 2-hour CZA model.

2. The authors discuss signal parameters throughout the paper (e.g., lines 88, 131, 146, 160, 216-218), with
 some details in the Supplementary Information about how these signal parameters are generated from
 extracted features of the nanomotion data. However, it is very difficult to understand what is actually being
 measured for each selected signal parameter; the examples listed in the Supplement are very difficult to
 interpret. More interpretable explanations for the users of what SPs were selected for each bug-drug pair
 would be helpful. If this is not possible because of complexity, and the intent is to view these SPs as a black
 box, this should be more explicitly stated as well. Assuming independent validation (see Major Critique 1),
 intuitively understandable SPs are not a requirement for a useful classifier, but especially at this early stage
 with limited testing, if SPs are explicable in some manner that relates to drug response, that might increase
 confidence in their generalizability. The attempt on lines 216-218 (“The SP in question for this model was
 related to the time dependence of the local minimum in the power spectrum estimated at the beginning and
 end of the drug phase”) is quite confusing.

*We agree that the description of signal parameters (SPs) needed to be improved. In general, a SP concrete*
 *mathematical value extracted from a transformed nanomotion signal. In a n agnostic approach (i.e., “black*
 *box”) several thousand SPs are calculated in the first step of machine learning and only the selection*
 *algorithm selected informative SPs in terms of antibiotic susceptibility. The final model (those presented in*
 *the text) are pareto optimal models, i.e., neither more nor less SPs will generate a better performing model.*
 *We undertook the following modifications to the manuscript to make the SP concept more understandable:*

1) *We rewrote the result section dealing with the algorithm development “**Classification algorithm***
 *development using machine learning based on SP” hoping to make the process more understandable*

and the SP concept more conceivable (lines 170-198). It mentions the origin and different kinds of SPs
(PSD, Quantile and MF DFA) that are further explained in the Suppl. Information. It also explains the
model training and selection process.

2) In general, we refrain to explain each SP used in the mathematical models in the main text (but in Suppl.
Info, see point 3) – in the new result section “**First general nanomotion AST for CRO and CIP**” (line
200) we explain exemplary SP1 for the CRO model.

3) We improved readability and extended the description in the Supplementary Information. We introduced
results for random signals for which SPs based on the quantile spectrum and MF DFA can detect changes
missed by power spectral density or variance. We also created descriptions of SPs used in each model
of the manuscript in the tabular form in Tables 1 to 6 in Supplementary Information.

4) We give a few examples of SPs already in the Introduction in line 95 and following.

5) We refer to previous studies (Ref 23, 32, 34) in which the ratio between the median variance between
drug and medium phase was calculated to determine susceptibility. This ratio can also be considered a
simple empirically found SP, see line 122 and following – as it is a mathematical value calculated from
the nanomotion signal. We also emphasize that the slope of the variance is one of these empirically
found SPs (line 142). However, through ML more informative SPs are selected.

3. The use of a single antibiotic concentration so far from the clinical breakpoint, chosen from the responses
of only two strains, to classify susceptibility is puzzling, particularly in light of the lack of a clear
relationship between their SPs and MIC (lines 141-150). And while it seems they must have determined the
optimal concentration for each bacteria-antibiotic combination, it is only discussed for ceftriaxone – a
similar figure to Fig 2b, used to determine each dose for each pathogen, might be added for other pairs as a
Supplemental Figure. Do the authors envision using a single dose for each antibiotic, for all pathogens? Or
might pathogen identity impacts the exposure dose? Independent concentration optimization on *K.*
*pneumoniae* could help verify the approach, given the similarity between the two species.

1) The manuscript shows the dose response for *E. coli* reference strains + CRO in Figure 2b using the
exponential slope (We updated this panel, as we accidentally showed the linear slope in the original
version. However, both fits indicated the same ideal concentrations for discrimination of S and R).
Figure 5b shows the dose response for ceftazidime-avibactam (CZA) also in the original version. In the
revised version, we added the Extended Figure 4 with dose responses for *K. pneumoniae* + CIP, *E. coli*
+ CIP and *E. coli* + CTX. For each of them we used, when possible, the ATCC reference strains. In the
case of *K. pneumoniae* R, we used a clinical isolate from the Swiss Reference Center for AMR, NARA
as we did not possess a *K. pneumoniae* ATCC MDR strain. Indeed, the response to CIP differed between
*K. pneumoniae* and *E. coli*, with *K. pneumoniae* being easier to discriminate using the exponential slope
and we have not been successful in creating a combined model for *E. coli* and *K. pneumoniae* with the
current set of SPs. However, a combined model proved successful for ceftriaxone and ceftazidime-
avibactam. We gained limited experience with measuring at two concentrations for another last-resort
antibiotic, not surprisingly this yielded very high performance, and could allow the development of
models classifying into three categories (S, R and extended exposure), very important for certain drugs
in terms of clinical decision making. We are working on a separate manuscript for that.

2) The reviewer points out that we use in nanomotion-based ASTs concentrations far from the clinical
breakpoints that were established for growth-based ASTs reporting MICs. We acknowledge the
apparent misalignment, but we can provide the following reasons for favoring higher concentrations:

a) We want to see the effects of the antibiotic fast as the main objective of the nanomotion AST is to beat
the conventional ASTs in the time to result. The higher the concentration the faster the kill rate (at least
for many antibiotics) - however if the concentration is too high prediction of false resistance (false
negatives) rise. The ideal concentration is a balance between these two factors. In the nanomotion AST,
we determined this concentration through dose-response experiments and the ideal concentration is
where the reference resistant strain (R) exhibits an increase in variance, while the susceptible strain
does not (Figure 2b, Extended Figure 4) References 23, 30, 32 and 38 also refer to previous publications
that use higher than breakpoint concentrations (lines 113-115).

b) MIC is a growth-dependent measure and as such not particularly useful for growth-independent
measurements such as nanomotions. Non-growth does not equal killing (MBC). Also, the MBC is not
particularly informative as it is also assessed by CFU assays, which depend on growth. In addition, the
killing can be slow and our measurement time. The physiological situation we observe with nanomotions
is a state in which bacteria might still be functional (exhibit metabolic activity) besides an inability to
divide and produce a culture or colony – comparing just the variance at MIC and media control supports
this (lines 145-149). Previous publications on *M. tuberculosis* support the growth independence. Here,
antibiotic susceptibility was measured at room temperature. At this temperature MTB cannot divide, yet
we could distinguish S and R phenotypes using high concentrations of antibiotics
(doi:10.1016/j.micinf.2023.105151 and reference 36). Higher concentrations of antibiotics affect the
metabolism again faster and were therefore chosen.

c) Higher antibiotic concentrations can be found in other phenotypic AST methods, as filter discs used in
the standard disk diffusion assays that use a single concentration much higher than the clinical BP (in
fact very similar to ours, e.g., 5 µg/ml CIP and 30 µg/ml CRO according to EUCAST regulations) to see
the effect on the bacterial population. Though it is to build a sufficient inhibitory gradient on the agar
plate, the initial idea in the experimental set up to work with concentrations above the BP is not new or
restricted to a nanomotion-based AST.

150 d) The development and adoption of MIC and MBC values as measures of AST date back to the 1950s (See,
https://doi.org/10.1093/jac/48.suppl_1.1), when no other but growth-based methods were available in
microbiology. Gene-based techniques used in the clinical microbiology labs, such as PCR, now
commonly used in diagnostics to detect carbapenemases or ESBLs, also required adoption of new “end
points”, i.e. presence or absence of a given gene. New technologies such as nanomotion will also require
the introduction of new “units” beyond MIC and MBC.

4. Lines 195-202: not clear how helpful more training with isolates near the breakpoint MIC will actually
be for classification accuracy, given the “non-intuitive” relationship between SPs and MIC (lines 141-150)
– it seems that making fine distinctions around the breakpoint would require a predictable relationship
between SPs and MIC. It is also not clear what the authors mean by the last sentence on line 205 in this
context – what “additional information for fast clinical decision-making” would be added by inaccurate
classification near the breakpoint MIC? Or is this referring simply to the accuracy for isolates farther from
the breakpoint MIC?

We agree that this needs a better explanation. SPs can be considered a certain aspect of the nanomotion
signal at a certain time interval. At this point this aspect is a solely physical observation and cannot be

*linked to a concrete metabolic or physiological process. The ML algorithm selects in an unbiased way SPs*
*that give the best performance in discriminating S and R strains (lines 174-182). It is well established that*
*different MICs also relate to different resistance mechanisms and very likely different nanomotions. By*
*incorporating strains with low resistance (low MIC) and (therefore most likely) different resistance*
*mechanisms into the training set, the algorithm will gain insights into various aspects of resistance*
*reactions. This approach ensures that the algorithm learns to categorize these strains more accurately,*
*enhancing the predictability of susceptibility and resistance around the breakpoints.*

*We also need to acknowledge that MICs are a very crude measure of the sum of underlying cellular*
*processes that lead to division or non-division. MICs are clinically applied because of the simplicity to*
*measure them. However, the underlying causes can be very diverse. Already considering morphological*
*changes as for instance cell elongation or cell lysis will both report susceptibility (no growth on plate) and*
*perhaps even report the same MIC all the while the underlying processes are different. While it is hard to*
*correlate elongation or lysis directly to an MIC value it can be relatively easy classified as S (similar to our*
*nanomotion method). In addition, MIC dependent breakpoints are frequently updated and hence are*
*removed from ISO 20776-2 (2021).*

*Concerning the Reviewers concern how clinical decision making can be improved by falsely classified*
*samples around the breakpoints (previously line 205): We removed the sentence about clinical decision*
*making. Our original intention was to emphasize that reporting MICs around the breakpoint is far from*
*flawless (slight changes in incubation, operator, state of the cells etc etc all affect MICs) while other*
*measures could support decision making - similar to molecular PCR tests used for carbapenemase detection*
*in carbapenem susceptible strains - which will lead in most cases to treatment escalation to antibiotics*
*other than carbapenems or carbapenem-inhibitor combinations. However, we agree that this is too*
*speculative at this point and confuses the reader.*

**MINOR COMMENTS:**

1. Lines 163-165: the “score” referred to here, and plotted in Fig 3-4, is not clearly described in the
Supplement. Given its use in main figures, a description of this score is warranted in the main text.

*The score is a weighted sum of signal parameters and offset. The weights and the offset are also fitted*
*parameters in the logistic regression algorithm. A negative score means that the strain is resistant and a*
*positive score means susceptible. We added the explanation in lines 183-187. Please also refer to the section*
*“Classification Model based on SPs” line 209 in Suppl. Information.*

2. Lines 165-167: From a clinical perspective, lumping “susceptible with increased exposure” together with
“susceptible” is fraught, since dosing would be different for the two states (and this is felt to be important
for clinical outcomes), yet presumably this assay as currently configured would not make this distinction.

*We work with a two-class model, therefore we cannot discriminate between three classes in the current set*
*up. We work with clinicians together and the benefit of a rapid AST is a fast actionable result where the*
*double-dose of an antibiotic could be administered (i.e., susceptible increased exposure) until reception of*
*a slower MIC reporting AST (presumably within the next few days) to potentially further de-escalate the*
*treatment to the normal dose. Cephalosporins are well tolerated and in fact susceptible with increased*

*exposure category isolates we haven't encountered in the NANO-RAST study (so far 248 patients,*
*manuscript in preparation).*

3. Lines 182-183: handling isolates tested multiple times does require some special consideration, but it is
not clear that using the median SP value from multiple measurements is appropriate – if the test is not to be
deployed on median measurements, then using them here for accuracy reporting seems questionable.
Instead, perhaps one test per isolate could be selected at random for overall accuracy reporting, whereas the
classification of multiple replicates of the same isolate could be separately tabulated as a measure of assay
precision?

*An SP is a parameter derived from the signal and they are used to build a model. The score is the*
*classification result obtained with such a model. We calculate a score for each recording separately.*
*Afterwards, we combine recordings per PBC sample and report the median score (lines 206-207). Positive*
*values predict susceptibility and negative values predict resistance as shown in Fig 3, 4, 6 and Extended*
*Figure 7. In general, working with replicates is standard in microbiology. It improves the accuracy. Also*
*in the clinical performance evaluation, i.e., Phenotech-1, we use triplicate recording per PBC sample.*

4. Line 188: For CRO testing, 30% of the experiments apparently used just two isolates (the reference S &
R E coli) – were these overrepresented in the training sets, or was only the median used? Was this imbalance
true for all bacteria-antibiotic combinations?

*The over-representation of two reference strains (ATCC-25922 and BAA-2452) in CRO model has*
*historical reasons. Once we started the work on model training, we did not have access to the number of*
*MDR strains that we had for the later models. We did not remove these additional recordings from the CRO*
*training data set, as ML becomes more reliable, and performance improves with larger data. This proved*
*right, given the superior performance for the CRO model that it achieved on the test data sets including the*
*clinical performance evaluation Phenotech-1 (see Fig. 4e). The next models were developed on more*
*balanced data sets. Please, see Suppl. Table 2 for the comprehensive list of strains and samples used.*

*Please also compare with the previous response.*

5. Figure 3 and elsewhere: “false positive” and “false negative” for classification are unclear (does
“positive” refer to resistant or susceptible?); “Major Errors” and “Very Major Errors” are more standard
terms for AST.

*We adhere to the ISO 20776-2 (2021) in which the older system of CA and MD, VMD and mD is replaced*
*by accuracy, sensitivity and specificity. We apologize for the oversight and added to the text the definitions*
*of TP, TN, FP, FN. A positive refers to susceptibility as it is a susceptibility test (also in line with the*
*ISO20776(2) 2021). The information was added to the legend of Figure 3c when the term appeared for the*
*first time.*

**Reviewer #2 (Remarks to the Author):**

The article by Sturm et al. proposes the use of machine learning and nanomotion data acquired from an
AFM cantilever for classifying resistant from susceptible bacteria. The measurements and data processing
are carried out on a library of clinical isolates of E. coli and Klebsiella Pneumonia, and suggest that the
sensitivity analysis can be carried out using machine learning with good accuracy.

Although I admire the large amount of data analysis and experiments performed by the authors,
unfortunately, the organization of the article, its structure and the findings does not have enough merit to
justify its publication in Nature Communications. Our reasoning is listed below.

1. Undoubtedly nanomotion technology has the potential to become an important player in the future of
Antibiotic Susceptibility Testing (AST) market. The use of AFM cantilevers for this technology, as authors
have used, is known for years and has been already highlighted in several review articles. Therefore, the
experimental part of the article is not new, and authors do not bring new advancements on that front. As a
result, many of the statements in the introduction and conclusion sections related to nanomotion technology
are already pretty well-known facts and neither challenges our current understanding nor pushes it to the
next level. The machine learning part is also not new as authors seem to use classifying algorithms (though
unclear what are these classifiers- which I reflect on later). Besides, the combination of both, nanomotion
and machine learning, does not seem to be a new idea either. In fact, the same group published this idea in
a similar article (see: Microbes and Infection (2023): 105151). Therefore, the present article does not seem
to further advance our experimental understanding of nanomotion technology nor offers new insights.

*We thank the Reviewer for the careful review of our manuscript. However, we strongly believe that our*
*work brings a significant advancement in nanomotion AST.*

*The main objective of this article is to describe a novel and very rapid (4 hours and 2 hours) AST that can*
*reliably predict susceptibility and resistance of a wide range of clinical isolates. We show the response of*
*364 E. coli and K. pneumoniae isolates to four clinically important antibiotics and use those for training*
*models. As requested by all Reviewers, we improved the manuscript with additional test data sets. CRO,*
*CIP and CZA models were tested on 91, 86 and 43 independent samples, and achieved 98.9 %, 89.5 and*
*93% accuracy, respectively (See Fig. 4cde and 6bc). We tested these models on a total of 305 samples with*
*strains non-identical to those used for training of the corresponding model. In addition, the CRO model is*
*currently being tested on patients' samples in the clinical trial Phenotech-1 (ClinicalTrials.gov ID*
*NCT05613322).*

*During the preparation of the revised version, we completed the first Interim Analysis of the Phenotech-1*
*clinical performance evaluation and decided to include the clinical data for the first 85 patients in the*
*manuscript. The CRO model achieved 97.6 % accuracy with TTR of 4 hours in this study. We believe that*
*this additional data is convincing and dispel doubts regarding the advancements we achieved, the quality*
*of our classification models and the clinical relevance of our work.*

*We learned from the Reviewers that the novelty of our machine learning process was difficult to understand*
*for the reader non-familiar with this field, therefore we describe the process in more detail in the manuscript*
*(line 170-198) and in the new version of the Supplementary Information. We transparently describe the*
*means and development needed to accomplish this by introducing the concept of signal parameters (SP),*
*the mathematical and statistical foundation of SPs (e.g. PSD, quantile spectrum, MFDFA, lines 172-173),*

*SP extraction and selection algorithms to obtain classification models and to validate them on a very diverse*
*set of clinical isolates.*

*The generalizability of the nanomotion-based AST was the real challenge up to now that no previous*
*publication could satisfactorily address. This study is the first to overcome this obstacle as evidenced in the*
*clinical data from the Phenotech-1 and other independent test sets (Fig. 4cde, 6dc).*

*A second aspect of this study is the information density entailed in the nanomotion signal that is only*
*partially understood and can be unearthed by ML. Previous publications rely solely on variance analysis*
*which is clearly insufficient to reliably classify susceptibility and resistance in clinical settings.* *To*
*demonstrate that, we added a new Extended Figure 3. In this Figure we unsuccessfully attempted to classify*
*1485 recordings performed with 160 E. coli and K. pneumoniae isolates (susceptible or resistant to CRO)*
*based on variance ratios as introduced in reference 23. Variance analysis works only on small and uniform*
*datasets and neglects additional information in the nanomotion signal. In fact, future work will be needed*
*to grow our knowledge of this phenomenon. None of the previous publications, except ours on MTB, touches*
*the information density entailed in bacterial vibrations.*

*The previous publication the Reviewer 2 is referring to, focuses on an AST for slowly growing M.*
*tuberculosis. The methodology applied in this work was significantly different and it even required the*
*adaptation of the device for BSL3 and work under laminar flow. Therefore completely different aspects of*
*the nanomotion signal were used for the analysis, resulting in different **informative SPs**. This is caused by*
*(i) the very different physiology of MTB compared to Enterobacterales, (ii) the recording length >20 hours,*
*(iii) different antibiotics with different MoAs were used (isoniazid and rifampicin). (iv) performance of MTB*
*algorithms was not tested in clinical settings on prospective patients' samples.*

*We see neither a conflicting overlap to the mentioned publication nor can a reader of the MTB publication*
*anticipate or recreate the finding of the current manuscript.*

2. Following up on the use of machine learning, it is unclear what type of classifier has been used and why.
This is at least not clear from the main text, and the signal analysis part in the SI is prepared in such a sub-
optimal fashion that it makes it difficult to follow. Only in several instances in the text authors mention that
the models used were all Pareto optimal, while it remains ambiguous how did authors find this optimality?
What hyper-parameters in what classification algorithms they were changing to find the Pareto optimal
condition. As a result, the reader remains confused how this classification and analysis is performed.

*We agree that the description of ML algorithm development was difficult to comprehend. We believe we*
*significantly improved the algorithm description in the revised version of the manuscript and*
*Supplementary information. Please see the section of Machine Learning in the main text (lines 170-198)*
*that is supposed to highlight the key steps in the ML process. Since the actual development requires a deeper*
*understanding of signal processing and physics more information is found in the Supplementary*
*Information and the Material and Methods section (also due to word limitations).*

*In brief, we used a logistic regression algorithm as a classifier. The input variables for this classifier are*
*SPs selected by feature selection algorithm. Pareto optimality is related to multicriterial optimization. We*
*defined two criteria: maximized classifier accuracy and minimized the number of SPs. We utilized forward*

*selection followed by backward selection with recording only Pareto optimal models. Model accuracy is*
*estimated by 300 times repeated 3-fold cross-validation. The process results in models with a few SPs. It*
*also means that the number of fitted model parameters is small in comparison to the number of training*
*and testing examples.*

3. Presumably, the only new element of the article is the use of, what authors call “signal parameters” (SP)
instead of variance of a signal as a metric to perform sensitivity analysis. The authors rationale in coming
up with these SPs is that the slope of the variance does not allow them to distinguish the sensitivity of
phenotypes across different concentrations. Thus, they come up with new metrics/SPs. Unfortunately, these
SPs are not even mentioned once in the whole manuscript, although they seem to be the main new thing the
article is offering. The SPs are brought in a very disorganized fashion in the SI, making it difficult for the
reader to identify at the end how many SPs and what has been used in authors classification. For instance,
looking at Fig 3b, where authors show improvement in performance by adding more SPs, it is not clear
what parameter is SP1, or what parameters authors considered for SP=3 or SP=4, and why for instance
SP=5 is not there in the figure and what this SP class really entails?

*We believe the description of SPs has significantly improved. Please see the text (Lines 170-198) and*
*Supplementary Information and our previous answer. We added a few examples of easy to comprehend SP*
*in the introduction (line 95 and following.) and used this term for SPs like variance ratios or the slope of*
*the variance etc in the result parts (lines 122-126, 142 and following).*

*In addition, we described all “more complicated” SPs used in all algorithms throughout the manuscript*
*(Tables 1-6 Suppl. Info). For the Reviewers concern regarding Fig 3b, the SPs are listed in the Suppl*
*information, Table 1. A 5th SP is omitted as the model had reached Pareto optimality with 4 SPs (please*
*see the manuscript text (lines 195-198) and previous response). The signal parameters are indexed*
*according to their importance. For instance, a model with 3 SPs contains all parameters of a model with 2*
*SPs and an additional new parameter. The new parameter is less important than 2 others, so its index is 3.*
*That is, the signal parameters of a model with 2 SPs are a subset of the SPs of a model with 3 SPs (lines*
*195-198).*

*For the CRO model we describe the most impactful SP1 in the text (lines 213-215, Table 2 in Suppl. Table*
*for all SPs used in the model) – all SPs in each model are described in Tables 1-6 in the Suppl. Information.*

*We hope that this additional information is satisfactory and provides enough insight into our methodology.*

4. Regarding the SP matter, it is also not clear how these SPs change before and after adding antibiotic.
There is only one schematic in the SI that to some extent defines SPs – But variation of SPs for real data
are completely absent, and thus their sensitivity could be a matter of question. For instance, the dip that can
be seen in PSD has been suggested to serve as an SP. But, the system under investigation is pretty noisy,
there is noise in amplitude and frequency, therefore, it is unclear how accurate such a SP could be used in
performing the analysis. Besides, it could be that this SP is fully masked under the nanomotion noise and
be strain dependent.

We believe there is a misunderstanding because of the absence of more detailed information regarding our
methodology. The Reviewer 2 states: “it is also not clear how these SPs change before and after adding
antibiotic.” We would like to clarify that the SPs do not change before and after adding antibiotics. SPs
should not be confused with signal estimators like the variance over time or flicker noise over time. For
instance, the dip in the PSD can also be considered a signal estimator over time not an SP.

SPs are constant parameters calculated from different time intervals. A SP is thus a mathematical value
representing an aspect of the nanomotion signal at a certain time. Calculating an SP using information
from the dip in the PSD is per se legitimate and the ML algorithm will choose it if it contains useful
information. The tables in Suppl. Info shows that the dip in the PSD was not selected for any model, i.e. it
was not used to classify S or R isolates. Our approach is agnostic, we do not intentionally exclude a feature
from the signal.

Regarding the comment: “it could be that this SP is fully masked under the nanomotion noise and be strain
dependent”; we agree that we work with the noisy system, that’s why Machine Learning is applied. ML
selects the best discriminating SPs to increase classification accuracy. They do not only convey the
information about the cell’s vibrations but also about the background noise. The process is similar to
building classifiers used in image analysis / visual inspection. If one can define a clear background,
knowing the light source and having an optimally calibrated camera, one can focus on object defects. But
in surveillance systems to detect intruder faces, classifier needs to be built on photos with different
backgrounds, time of day, season, etc. Our ML process is also summarized in Extended Figure 1.

5. Following up on the same, for instance two frequencies around cantilever resonance are chosen as SPs. I
doubt such SPs could be a good choice as they very much depend on the experimental conditions and the
cantilever properties. It could be very well that there are slight changes in the stiffness/mass of the
cantilevers in different experiments which could lead to different SPs and could easily influence the
classification procedure. Even if we assume that the cantilevers in all experiments were closely identical,
the initial cell density/mass is highly varying based on Extended Fig 2. This becomes even more important
considering that the resonance peak and the frequencies nearby remain unaffected in the presence of
antibiotics in nanomotion experiments: see for instance Fig. 3 in <http://dx.doi.org/10.1063/1.4895132>,
which clearly supports points 4 and 5.

We believe that in the absence of more detailed information regarding the SPs used, the Reviewer 2
misunderstood the use of SPs.

The statements:

“two frequencies around cantilever resonance are chosen as SPs.” is not true.
“SPs (...) very much depend on the experimental conditions and the cantilever.”

are not true.

Please see Tables 1-6 in Suppl. Info for detailed SPs description.

*The diagnostic environment is quite different from a clean scientific lab. The cantilever functionalization*
*procedure is not repeated multiple times to achieve optimal conditions. Measurements are not done in the*
*basement with optimal vibration isolation. Everything happens on the diagnostic table with other diagnostic*
*devices working around and multiple people present. For this reason, the noise background is much richer*
*and cannot be ignored.*

*These SPs indeed reflect mechanical properties of the sensor and cells attached. The separation signal from*
*noise is achieved in the training process and is often standardized by calculation of the ratios between SPs*
*that makes them relative one to another. For this reason, multiple SPs are calculated and only optimal ones*
*are selected by ML.*

6. As a result, likely most of the SPs chosen by the authors are independent of the nanomotion signal itself
and do not seem to have clear correlation with the biological fingerprints. Therefore, other SPs shall be used
for classification purposes that are biologically relevant. One such SP, that apparently authors also
considered (but again it is not clearly mentioned) is the slope of the spectrum. However, it is pretty well
established that the PSD of E. coli is characterized by $1/f^a$ where a has been obtained independently by
different groups to be close to 2, and that the spectrum becomes white if the bacteria are dead. Therefore, it
is also not clear what new information such a metric/SP can offer other than what can already be entailed
from variance.

*The Reviewer 2 states that: “SPs chosen by the authors are independent of the nanomotion signal itself and*
*do not seem to have clear correlation with the biological fingerprints.”*

*If we may use the example of MALDI TOF widely used these days in the diagnostic labs for species ID. It*
*uses agnostically information from the mass spectrum without knowing the biological meaning behind each*
*peak to train models for species ID. It is safe to assume that these peaks however correlate with a biological*
*meaning - though it's not known. Like Maldi-tof and species ID, nanomotion and AST contains biologically*
*relevant information even though it is not explicable at this point. In general, our approach is agnostic, and*
*we do not limit the search for SPs to any known biological relevance, simply because barely anything is*
*known at this point about which biological processes relate to which part of the nanomotion signal. We*
*added this to the main text (lines 178-182):*

*Furthermore, we do not use a slope inside the spectrum, as Reviewer 2 suggests, but the slope of the*
*variance over time as they are depicted in Figure 1d, 2a, 2c and 5a. Slopes inside the PSD are signal*
*estimators and used to calculate spectral SPs. Our ML algorithm can select from them if they are*
*informative in separating R from S phenotypes (please compare Tables in Suppl. Info). However, the only*
*spectral SPs used are integrals of PSD for the CRO algorithm (Suppl. Information table 2)*

Also, few more unclarities related to the experiments and sample preparation:

• The authors compare susceptibility data obtained by EUCAST conditions to experimental conditions but
performed their own experiments in 50% LB medium + 50% deionized water, which is very different from
EUCAST conditions. The growth rate and doubling times of microorganisms might be very different in
their case versus EUCAST. In their earlier study (Nature Nano, 2013) the authors claim that the nanomotion
couples with the growth rate of the microorganisms.

*The nanomotion AST is a growth-independent method. We do not measure microbial growth rate on the*
*cantilever. We compare our results to growth-based disc-diffusion and E-test because they are the reference*
*methods acc. to CLSI and EUCAST*

*We agree that nanomotions are affected by growth on the cantilever or in the medium, by the increase of*
*biomass for instance. Yet, growth cannot be equaled to an increase in the variance and, importantly,*
*nanomotions also happen without growth. In addition, we did not observe major differences between LB*
*and MH during nanomotion experiments (here simply variance analysis), in fact the media are not that*
*different in their overabundance and quality of nutrients.*

*We have another manuscript under revision using the bacterial toxin system MazEF to enforce a non-*
*growing state of E. coli cells. Nanomotions still occur - we could share this document if the Reviewer 2*
*wishes. In addition, the MTB publication from our group earlier this year measures at room temperature*
*in which MTB is not growing (reference 36).*

• Why is there such a high variation in the nanomotion variance (and often a clear upward trend) during the
2h measurement even without the addition of the antibiotic? (see eg. Fig1d and fig.2c , fig 5a, fig. s3).

*The increase in variance is most likely due to a subtle increase in the biomass (we measure at room*
*temperature in culture media, i.e., we do not prevent reproduction in the medium phase) and second, the*
*bacterial population on the cantilever is adapting to a new environment after being cultured in a different*
*media (blood culture) and a floating lifestyle. Therefore, we expect a lag or adaptation phase. We speculate*
*that this is quite heterogeneous among the population and could explain the noise. It is the primary reason*
*for a 2-hour medium phase. At 37°C the medium phase can be shortened to 30 min.*

• It is unclear why did the authors use a high concentration of the CRO antibiotic (32ug/ml) , considering
the breakpoint is at 1ug/ml?

*We elaborate on this issue in the response 3 to Reviewer 1.*

• What is the role of gelling agent, agarose, in the sample preparation? It is not mentioned in the article.

*We determined empirically that a gelling agent stabilized the bacterial attachment and led to a more even*
*distribution. We agree that the reader should be informed of the reason why we added it. For that we added*
*to Figure 1c legend: “The gelling agent proved beneficial for an even distribution and stability of the*
*bacterial attachment.”*

• Table 2 is missing even though it is mentioned in the text.

*We cannot find the reference to Table 2. We only refer to Supplementary Tables. We now rephrased them*
*to Extended Tables. Extended Table 1 lists all strains used in the study, Extended Table 2 lists all*
*experiments and their scores for their corresponding models.*

For all the above, I do not recommend this article for publication in Nature Communications.

**Reviewer #3 (Remarks to the Author):**

In the manuscript titled “Accurate and rapid antibiotic susceptibility assessment using a machine learning-
guided nanomotion technology platform,” the authors present an integrated nanomotion technology
platform for measuring bacterial response to antibiotics. Nanomotion technology, based on atomic force
microscopy, provides a growth-independent approach to assessing bacterial response. The platform
combines hardware and software components, enabling fast and reliable analysis of large datasets. Machine
learning algorithms were employed to accurately classify antibiotic-resistant and -susceptible bacteria. The
study focuses on testing the response of *E. coli* and *K. pneumoniae* strains to four commonly used antibiotics
for bloodstream infections.

The nanomotion technology platform presented in the study offers a novel approach to antibiotic
susceptibility testing that overcomes the reliance on bacterial growth assessment. As clearly described by
the authors, it demonstrates faster results compared to existing methodologies, with a turnaround time of
four hours instead of 24 hours. The platform eliminates the need for plating and enables a quicker overall
process for obtaining patient antibiograms. Importantly, it achieves high classification accuracy rates
(90.5% to 100%) for fluoroquinolones, cephalosporins, and cephalosporin-inhibitor combinations,
comparable to standard clinical diagnostic methods.

In summary, the manuscript is well-written, with very clear figures, and I recommend its publication without
any reservations. However, I have a few minor suggestions and questions for clarification that could
enhance the manuscript's clarity and impact.

1. In the results session, regarding the nanomotion recordings:

a. It would be beneficial to provide more information on the specific resistance mechanisms that may be
causing the observed increase in signal variance at sub-MIC concentrations of CRO. This clarification
would help readers understand the underlying biological processes contributing to the nanomotion signals.

*We process the nanomotion signal to develop ML models capturing general information about resistant*
*and susceptible isolates, independent of specific resistance mechanisms. Detailed translation of resistance*
*mechanisms to the nanomotion signal is limited due to insufficient data on resistance mechanisms of each*
*isolate used for training. We have added the Extended Figure 6 for *E.coli/K. pneumoniae* + CRO variance*
*plots, showing diverse responses of isolates for which we knew the kind of beta-lactamases - however the*
*responses are likely influenced by factors beyond resistance mechanisms (except for the ATCC strains none*
*of the strains is sequenced and were obtained from different sources and suggest high genetic diversity).*
*We added a short note to the text also referring to Extended Figure 6 (lines 165-168)*

*Future research is necessary to explore the complexity of the raw signal for delineating resistance*
*mechanisms.*

b. Please further explain the reasons behind the lack of correlation between the nanomotion signals and
MIC values. It would be valuable to discuss potential factors or variables that could influence the
nanomotion signals independently of the MIC values.

*We have partly addressed this concern in the replies to previous Reviewers' comments. Please see Response*
*to Reviewer's 1 major comment 3) point 1 and point 2bc (lines 107 and following, lines 125 and following*
*in this document); as well as major comment 4) lines: 164 and following in this document.*

*MIC is simply a measure of the absence of growth. A bacterium is impaired in cell division but beyond that*
*exhibit cellular functionalities/metabolic activity. The MBC is a better measure but barely used in any*
*diagnostic (because of labor intensity and slow TTR) - however, the MBC is usually determined in CFU*
*assays and therefore the assay itself relies on growth and does not capture the state of a cell in which it is*
*still alive but has lost its ability to divide – we grasp this state through vibrations.*

5151) *The nanomotion AST is more sensitive compared to growth-based methods. The physiological situation we*
*observe with nanomotions is a state in which bacteria might still be functional (exhibit metabolic activity)*
*besides an inability to divide and produce a culture or colony. lines 145-149 and reference 23 as well as*
*lines 163-165 – references 39, 42-44.*

*In future research we might find a correlation for both, MIC and MBC, to SPs in the nanomotion signal.*
*For that purpose, however, regression algorithms need to be developed that require ML with many more*
*data points for each MIC.*

*Regarding classification models: The signal parameters chosen by ML algorithms are extracted from the*
*drug phase and describe therefore an aspect of the reaction to the antibiotic. We do not know what this*
*information in the SP biologically relates to at this point (which will include a massive effort of time and*
*resources to identify and is beyond this study). The most intuitive SPs are spectral SPs for Ec/Kp + CRO.*
*SPI for instance describes the ratio between two time intervals: the integral of the PSD in the frequency*
*range 20-28 Hz of 90-120 min and 0-30 min of the drug phase. See lines 213 and following and Suppl.*
*Information Table 1.*

c. In order to ensure the generalizability of the nanomotion AST, it is important to address the diverse
responses observed among different strains and antibiotics. Please elaborate on your plans to tackle this
issue and provide details on any strategies you have in mind to account for the variability.

*This is a very relevant remark and the reason for this publication is to show that to translate nanomotion*
*AST into the clinic the vast diversity of responses to antibiotics needs to be addressed and the “simple”*
*analyses that were so far published are underperforming for (see Extended Figure 3 – variance ratio*
*analysis). For this reason, we acquire(d) strains from different geographies, and used 364 E. coli and K.*
*pneumoniae isolates to train the ML algorithms. In general, the more extensive the training the better is the*
*generalizability, this is what we also observe in the performance of the newly added independent test sets*
*for CRO, CIP and CZA. CRO is the most extensively trained model with the best test performance of an*
*accuracy of 97.6% in the Phenotech-1 study over three different hospital sites (1/3 of study population,*
*namely 85 patients, has been analysed and added to the manuscript, study protocol and Interim Analysis is*
*attached).*

*The second point is, and we added this to the discussion, the definition of new SPs. Signal processing is*
*continuing, and more general SPs will be identified during a learning process that will improve our*
*understanding of the nanomotion signals.*

*We foresee a continuous expansion of the nanomotion database, and models' retraining with new data*
*originating from a large number of isolates of hospitals from different geographies. The more popular the*
*nanomotion technology becomes, the better will be the classification algorithms, as it is being observed*
*with MALDI-TOF for strain ID.*

*In conclusion, the training of these models can be expanded by adding more strains and experiments but*
*also by improving and finding (or defining) new SPs conveying new aspects of the nanomotion signal.*

553 d. It would be helpful to provide more details on how the nanomotion signals correlate with bacterial
vibrations rather than growth. This explanation would enhance the understanding of the physical
phenomena underlying the nanomotion technology.

*This claim was addressed in the Nature Nanotechnology paper of 10.1038/nnano.2013.120, in which E.*
*coli cells were measured under nutrient depletion conditions (PBS) and the deflection of the cantilever was*
*well quantifiable (though understandably different to culture media). We cite this key publication at several*
*instances in the manuscript. Furthermore, we recently published 10.1016/j.micinf.2023.105151 a study on*
*the development for a Mycobacterium tuberculosis AST in which also S and R phenotypes of clinical MTB*
*isolates were delineated. Experiments were performed at room temperature at which MTB does not grow*
*yet we could generate algorithm with high performance based on SPs (though not identical with those in*
*this study). We conducted another study which is currently under revision for a study on E. coli expressing*
*low levels of the toxin mazF thought to be involved in persister formation. Here we show that E. coli*
*expressing mazF exhibited no growth but was measurable using nanomotion technology. Upon request, we*
*could share the current version of the latter manuscript, which is not yet publicly available.*

*We continue the research on the origin of nanomotion and we are aiming at publishing this work in the*
*peer review journal, however this is beyond the scope of this manuscript.*

2. In the results session, regarding the classification algorithm:

a. Please provide more details on the specific machine learning algorithms used for the development of the
classification models in the nanomotion AST. It would be beneficial to include this information in the main
manuscript rather than solely in the Supplementary Information, as it will facilitate comprehension for
readers.

*We modified the description of the algorithm in the manuscript in the result section "Classification*
*algorithm development using machine learning based on SPs" (lines 170-198). This section has been*
*rewritten. We also rewrote the Supplementary Information. We now present signal parameters in tabular*
*form (Suppl. Info Tables 1-6) and added results of illustrative simulation of random signals for which*
*proposed signal parameters outperforms basic variance calculations (see Extended Figure 3 for which S*
*and R almost entirely overlap when the ratio between variance in medium and drug phase are calculated*

*versus Figure 4 a – bot use the same dataset of 160 Ec and Kp isolates). We would also like to ask this*
*Reviewer to refer to our response to Reviewer 1 comment 2.*

b. How did you determine the optimal number of signal parameters (SPs) to extract from each nanomotion
recording? Please elaborate on the methodology used to determine the appropriate number of SPs and any
considerations taken into account during this process.

*We applied a feature selection algorithm that starts with forward selection followed by backward*
*elimination. The recorded models are selected on the basis of Pareto optimality criterion. In practice*
*forward selection is continued to the moment of lack of improvement for the accuracy. Models with lower*
*or the same accuracy but larger number of signal parameters are not Pareto optimal because the number*
*of signal parameters is the second criterion in the multicriterial optimization process. That is, the accuracy*
*is the first criterion and the number of signal parameters is the second one (accuracy is maximized, number*
*of signal parameters minimized). The same holds for backward elimination if removal of a signal parameter*
*increases accuracy the previous model is forgotten (because it is not Pareto optimal - lower accuracy and*
*larger number of signal parameters). Accuracy is estimated with 300 times repeated 3-fold cross-validation*
*procedures.*

*This information can be found lines 183-190, more details can be found in the M&M section “Machine*
*Learning and development of classification models” lines 805 and following. The Suppl. Info gives more*
*detail and background information as well as an overview of all SPs used in the study.*

c. Can you explain the process of repeatedly selecting SPs during the development of the classification
models? Please provide an overview of the methodology used and any criteria used to guide the selection
process.

*We believe we explained it in the previous response.*

603 d. Please elaborate on the scoring system introduced for the classification models and how it relates to
604 predicting susceptibility and resistance. A more detailed explanation of the scoring system will help readers
understand its practical implications and its connection to the prediction outcomes.

*The score is simply a weighted sum of signal parameters and offset. The weights and offset are fitted during*
*training of the logistic regression model. The positive score means strain is susceptible, the negative means*
*it is resistant. In fact, the logistic regression algorithm uses this sum as an input to a logistic function to*
*map score into values ranging from 0 to 1. The model parameters are fitted to align the output of the logistic*
*function with the labeled data from the training dataset.*

*The classification scores for all recordings are provided in Extended Table 2 – pls. see tabs for each model.*

3. In the results session, regarding the applicability of the methodology.

Please explain the rationale behind selecting only one signal parameter (SP) for the classification model to
discriminate susceptible and resistant strains of E. coli treated with ceftazidime-avibactam. Providing an
explanation for this specific choice will help readers understand the reasoning behind the decision and the
potential advantages or limitations associated with using a single SP in this context.

*The CZA 4-hour model was trained as any other model in this study, i.e., with the principles of Pareto*
*optimality. Since we achieved 100% accuracy with one SP no further SP needed to be added. In other*
*words, one SP carried sufficient information to discriminate all S from R phenotypes. We believe this is*
*because of the generally strong bactericidal effect of CZA (affecting all S rapidly) and the relatively uniform*
*resistance mechanisms - as CZA has only been introduced to the market in the late 2010s the time for*
*evolving diverse responses to CZA was limited - not comparable to CRO, CTX and CIP for which we know*
*several different resistance mechanisms.*

Overall, the manuscript is well-written and provides a valuable contribution to the field of antibiotic
susceptibility testing. Addressing these suggestions and providing further clarification will enhance the
manuscript's clarity and impact.

REVIEWER COMMENTS

Reviewer #1 (Remarks to the Author):

Sturm et al, "Accurate and rapid AST using a machine-learning assisted nanomotion technology platform", resubmission to Nature Communications

This manuscript discusses using cantilever-based bacterial nanomotion detection to test for susceptibility to four different antibiotics (from two different classes) in two different species of bacteria. They incorporate machine learning to identify signal parameters (SPs) that best distinguish susceptible from resistant isolates. The manuscript is greatly strengthened by the incorporation of an independent testing set of isolates, as the major concern in the initial submission was the risk of overfitting based on the number of potential SPs that vastly exceeded the number of isolates in the training set. The description of SPs, including explicit acknowledgment of their lack of direct connection to known biological parameters, is also improved.

A few remaining comments on the manuscript:

1. Authors should clarify explicitly that the SPs chosen (eg, the 4 SPs for the K pneumoniae + CIP model as shown in Fig 3), and the classifier model itself (ie the weightings of each SP, etc used in classification), were not altered at all before applying to the testing set. Lines 218-221 clarify that no strains were retested, which is important, but another critical point is that neither feature selection nor classifier training was repeated on the new isolates. This is implied as written, but should be explicitly stated.
2. Line 150: not clear that the dose that best discriminates between a very resistant and a very susceptible isolate, is necessarily the best way to choose a dose for all isolates – indeed, this may be at the core of their poor performance on isolates at borderline MICs? (see point 3)
3. Discussion of poor performance near the breakpoint MIC should include explicit note that independent testing sets include very few isolates near this breakpoint, including no resistant isolates near the breakpoint (Fig 4c-d) – this general issue is dealt with better in the Discussion (lines 294-296), where the suggestion of further training with different exposure doses makes good sense, than in the Results (lines 234-237), where this inaccurate classification near the breakpoints is claimed to “underscore the robustness of the method”, a claim that should probably be omitted or rewritten
4. Line 207, and authors' response to initial review: the use of median values to train the classifier and especially to assess performance is still fraught, despite the authors' claim in response to review that “working with replicates is standard in microbiology; it improves the accuracy”, unless the eventual assay also intends to test replicates and use median values (in which case this needs to be made explicit). Otherwise, assuming the eventual assay will test each strain only once, it is the results of individual tests, with any inherent noise, that will dictate ultimate assay performance, and using medians would reduce that noise in a way that would overstate ultimate assay performance. In the end, this is less critical because of the independent validation, although for CIP (unlike for CRO) it appears that replicates were once again used (more samples than strains). Here, individual test results should be reported, not medians, and perhaps even a report of assay concordance for strain replicates.

5. A few points to consider (or tone down) in the Discussion:

a. Line 267: “entirely new approach to AST” is an overstatement given the body of published work on this cantilever-type approach for AST

b. Line 270: re: bypassing of plating step, could the authors speculate on how mixed cultures would affect their assay? This happens in a substantial fraction of bacteremias, especially from GI or biliary sources (and is one reason why current workflows do not yet incorporate eg direct disk diffusion assays on blood culture broth, despite good overall concordance with subculture)

c. Line 282-283: given the context (discussing a diagnostic path to eliminating the need for empiric antibiotic use), it should be clarified that this assay still requires an initial liquid culture step from blood, ie that the 2-4 hour assay time follows an initial positive culture that typically takes 1-3 days (this is still an advance over the current workflow, but as written the text implies that with this assay, no empiric antibiotics would be needed, which is not the case)

Reviewer #2 (Remarks to the Author):

The authors have answered my concerns satisfactorily. The description of the ML algorithm implementation, that was previously unclear, is improved. The description of the signal parameters which seemingly was a comment from all reviewers has also improved.

With the current amendments, I have no objection against the publication of the work.

Response to Reviewer Comments

Reviewer #1 (Remarks to the Author):

Sturm et al, "Accurate and rapid AST using a machine-learning assisted nanomotion technology platform", resubmission to Nature Communications

This manuscript discusses using cantilever-based bacterial nanomotion detection to test for susceptibility to four different antibiotics (from two different classes) in two different species of bacteria. They incorporate machine learning to identify signal parameters (SPs) that best distinguish susceptible from resistant isolates. The manuscript is greatly strengthened by the incorporation of an independent testing set of isolates, as the major concern in the initial submission was the risk of overfitting based on the number of potential SPs that vastly exceeded the number of isolates in the training set. The description of SPs, including explicit acknowledgment of their lack of direct connection to known biological parameters, is also improved.

A few remaining comments on the manuscript:

1. Authors should clarify explicitly that the SPs chosen (eg, the 4 SPs for the K pneumoniae + CIP model as shown in Fig 3), and the classifier model itself (ie the weightings of each SP, etc used in classification), were not altered at all before applying to the testing set. Lines 218-221 clarify that no strains were retested, which is important, but another critical point is that neither feature selection nor classifier training was repeated on the new isolates. This is implied as written, but should be explicitly stated.

We agree that this point needed to be clarified. We added this statement to this section (lines 221-222): "Importantly, none of the isolates used for training were reused for testing and neither feature selection nor classifier training was repeated on the new isolates."

2. Line 150: not clear that the dose that best discriminates between a very resistant and a very susceptible isolate, is necessarily the best way to choose a dose for all isolates – indeed, this may be at the core of their poor performance on isolates at borderline MICs? (see point 3)

We utilized both reference strains for dose-response testing, mandated by ISO-20776 as quality assurance measures. The assay yielded valuable insights into the concentration that effectively differentiated between the two strains. Addressing a major critique point from the previous submission (no. 3) about the choice of concentrations – speed was an essential concern, i.e. seeing the response to the antibiotic asap. Given the current technological set-up allowing measurement at only one concentration, the dose response testing is a pragmatic compromise and results in classification models with excellent performance.. This is particularly noteworthy given the range of isolates, such as 0.01 - 10000 µg/ml in the case of CRO.

While it's acknowledged that low-resistant strains exhibit distinct behavior (lines 236-238, 296-298), the performance of dose response testing with a representative low MIC strain is difficult because representative strains are hardly available. On the other hand, increasing the number of low resistant strains for dose response testing is impossible due to the labor and cost-intensity. The dose response testing is a preparatory step and the use of a single SP "variance slope" is simplified but a sufficient approach to determine the concentration for the ultimate test. In contrast, the classification models, relying on 4-6 SPs, provide a more comprehensive depiction of varied responses, aligning with expectations for strains with diverse susceptibility levels.

In conclusion, it is speculative why the performance at low resistance levels is worse. It could be improved by optimizing the antibiotic concentration or training the models with more of these isolates (which are admittedly not very frequent) – or a combination of both. We think we have encompassed both possibilities at the two previously mentioned sections in the manuscript, lines 236-238 (regarding training with low resistant isolates) and 296-298 (measuring at different concentrations).

3. Discussion of poor performance near the breakpoint MIC should include explicit note that independent testing sets include very few isolates near this breakpoint, including no resistant isolates near the breakpoint (Fig 4c-d) – this general issue is dealt with better in the Discussion (lines 294-296), where the suggestion of further training with different exposure doses makes good sense, than in the Results (lines 234-237), where this inaccurate classification near the breakpoints is claimed to "underscore the robustness of the method", a claim that should probably be omitted or rewritten

We omitted this statement "underscore the robustness of the method" in the result section as suggested by the Reviewer. The authors however know that the MIC breakpoints are not perfect leading for instance to the intermediate category and the reporting of minor errors (in the classical way of reporting of AST performance (IS)-20776 (2007), not the new ISO-20776 (2021) which we believe is a big improvement on that matter). It might

very well be that antibiotic responses measured by nanomotion, and antibiotic responses measured by classical methods will not align perfectly - also in the future. Which one in the end correlates better with treatment success will be a subject of clinical studies. Again, nanomotion-based AST is not a growth-based assay, and we can only correlate our results to the MIC methods, i.e. the read out is different. The original point the authors wanted to make is that reference measurement results (MICs, diameters) around the breakpoints are frequently not repeatable in gold standard methods based on growth (Kirby Bauer etc.) – nanomotion results however were repeatable. Eventually, our dataset is small, and we agree with the reviewer to omit this part.

Lines 236-238 now read: “In the future, these models require training with more isolates exhibiting MICs around the breakpoints to better capture the nanomotion phenotype corresponding to the underlying low-resistance mechanisms.”

Furthermore, to address the first point in comment 3 of the Reviewer, we think all test sets and the distribution of MICs are transparently and explicitly shown in Figure 4c and d. Each isolate is listed in the supplementary tables and each experiment in supplementary table 2.

4. Line 207, and authors' response to initial review: the use of median values to train the classifier and especially to assess performance is still fraught, despite the authors' claim in response to review that “working with replicates is standard in microbiology; it improves the accuracy”, unless the eventual assay also intends to test replicates and use median values (in which case this needs to be made explicit). Otherwise, assuming the eventual assay will test each strain only once, it is the results of individual tests, with any inherent noise, that will dictate ultimate assay performance, and using medians would reduce that noise in a way that would overstate ultimate assay performance. In the end, this is less critical because of the independent validation, although for CIP (unlike for CRO) it appears that replicates were once again used (more samples than strains). Here, individual test results should be reported, not medians, and perhaps even a report of assay concordance for strain replicates.

We understand the Reviewers concern. In fact, the “eventual assay” is based on replicate measurements, and we reduce “noise” by using the median of usually three experiments (referred to as “recordings” in the text). The assay is under clinical investigation and uses triplicate measurements (NANO-RAST and PHENOTECH-1 studies). The Phenotech-1 performance of the first 85 samples were included in the result section (Fig. 4e ‘Phenotech-1 Test’) and indeed here we measure in triplicates and use a majority vote between replicates which mathematically corresponds to the median of the replicates. Given the often-poor repeatability in standard ASTs (see our comment to critique 3) closer to breakpoints measuring in replicates (which is not performed in the routine) would always be a plus.

We attached in the previous resubmission the interim analysis of the Phenotech-1 study. In Table 8 of this document the sample performance for *E. coli* +CRO is listed, which would be reported to the attending physician (if it wasn't a clinical performance evaluation, i.e. an observational study). These results are depicted in Table 4e in the manuscript (header: Phenotech-1 Test). Table 9 in the interim analysis lists the recording level. We see indeed an improvement in the performance which is why the final assay is performed in triplicates and final decisions are based on sample level. As the final assay is based on replicate measurements, we believe the presentation of the results in the manuscript are therefore accurate and justified.

We clarified this in the text to avoid a misconception.

Lines 207 and 208: “we obtained a score for each recording of a PBC sample and used the median score for sample classification and reporting of the final result.”

clinical performance (final assay), lines 225-228: “On a current sample size of 85 strains, including *E. coli* and *K. pneumoniae*, the model achieved an accuracy of 97.6% with a mean TTR of 4.24 h (SD = 0.21 h) across three different hospital sites (Fig. 4e). Here, results were again reported based on triplicate measurements.”

We also included a report for one of the samples of the Phenotech-1 study that emphasizes that the final test (as it is during the performance evaluation) is performed in triplicates. To uphold patient confidentiality, the report has been appropriately redacted. It is intended exclusively for the editor and reviewer's scrutiny.

5. A few points to consider (or tone down) in the Discussion:

a. Line 267: “entirely new approach to AST” is an overstatement given the body of published work on this cantilever-type approach for AST

We replaced it with “represents different approach to AST” now line 268

b. Line 270: re: bypassing of plating step, could the authors speculate on how mixed cultures would affect their assay? This happens in a substantial fraction of bacteremias, especially from GI or biliary sources (and is one reason why current workflows do not yet incorporate eg direct disk diffusion assays on blood culture broth, despite good overall concordance with subculture)

In the current setting we exclude polymicrobial samples coming from a patient sample (i.e., non-spiked PBC) as we cannot exclude the potential impact of a second or third species with a potentially different susceptibility phenotype. Very preliminary results coming from urine samples (which are arguably more affected by polymicrobial samples or contamination) suggest to a certain degree that this is tolerable (especially contaminations with Enterococcus spp.). Considering that a few hundred bacteria are sitting on the cantilever a few contaminating bacteria are unlikely to dominate the deflections of the cantilever. For blood cultures we do not have enough data to make a clear statement. In the worst-case scenario, as our assay commences after species ID (preferentially MALDI-TOF), it is known if a sample is polymicrobial and would unfortunately delay the process by using an isolate from plate.

In the current situation where we simply don't know its impact and to avoid speculations, we amended the sentence in lines 273-274: "...this method bypasses the plating step that usually occurs after PBC sample collection and before cartridge inoculation in current automated AST systems although polymicrobial samples were not tested in this study and could affect the results."

As a side note, we work with several hospitals and several of them start the AST directly from a positive blood culture (PBC) given a MALDI TOF protocol for species ID from a PBC is available as well. There are certain differences in accuracy for Gram+ and Gram- but the plating step between PBC and AST becomes less widespread in hospitals at least in Europe where MALDI-TOF is common, because it allows for a significant reduction of TTR and it benefits the patients.

c. Line 282-283: given the context (discussing a diagnostic path to eliminating the need for empiric antibiotic use), it should be clarified that this assay still requires an initial liquid culture step from blood, ie that the 2-4 hour assay time follows an initial positive culture that typically takes 1-3 days (this is still an advance over the current workflow, but as written the text implies that with this assay, no empiric antibiotics would be needed, which is not the case)

We clarified this section: lines 283-285: "A drastic reduction in AST TTR is necessary to limit the time of empirical drug administration and early switch to an informed decision-based paradigm. With the set-up outlined in this study, a nanomotion-based AST takes two or four hours starting from a PBC."

Reviewer #2 (Remarks to the Author):

The authors have answered my concerns satisfactorily. The description of the ML algorithm implementation, that was previously unclear, is improved. The description of the signal parameters which seemingly was a comment from all reviewers has also improved.

With the current amendments, I have no objection against the publication of the work.